# *NAT2* global landscape: Genetic diversity and acetylation statuses from a systematic review

**Jorge E. Gutiérrez-Virgen**[1], **Maricela Piña-Pozas**[2], **Esther A. Hernández-Tobías**[3], **Lucia Taja-Chayeb**[4], **Ma. de Lourdes López-González**[1], **Marco A. Meraz-Ríos**[5], **Rocío Gómez**[1]*

1 Departamento de Toxicología, CINVESTAV-IPN, Mexico City, Mexico, 2 Instituto Nacional de Salud Pública, Centro de Información para Decisiones en Salud Pública, Mexico City, Mexico, 3 Facultad de Salud Pública y Nutrición, Universidad Autónoma de Nuevo León, Monterrey, Nuevo León, Mexico, 4 División de Investigación Básica, Instituto Nacional de Cancerología, Mexico City, Mexico, 5 Departamento de Biomedicina Molecular, CINVESTAV-IPN, Mexico City, Mexico

* mrgomez@cinvestav.mx

**Data Availability Statement:** All relevant data are within the paper and its Supporting Information files.

## Abstract

Arylamine *N*-acetyltransferase 2 has been related to drug side effects and cancer susceptibility; its protein structure and acetylation capacity results from the polymorphism's arrays on the *NAT*2 gene. Absorption, distribution, metabolism, and excretion, cornerstones of the pharmacological effects, have shown diversity patterns across populations, ethnic groups, and even interethnic variation. Although the 1000 Genomes Project database has portrayed the global diversity of the *NAT*2 polymorphisms, several populations and ethnicities remain underrepresented, limiting the comprehensive picture of its variation. The *NAT*2 clinical entails require a detailed landscape of its striking diversity. This systematic review spans the genetic and acetylation patterns from 164 articles from October 1992 to October 2020. Descriptive studies and controls from observational studies expanded the *NAT*2 diversity landscape. Our study included 243 different populations and 101 ethnic minorities, and, for the first time, we presented the global patterns in the Middle Eastern populations. Europeans, including its derived populations, and East Asians have been the most studied genetic backgrounds. Contrary to the popular perception, Africans, Latinos and Native Americans have been significantly represented in recent years. *NAT*2*4, *5B, and *6A were the most frequent haplotypes globally. Nonetheless, the distribution of *5B and *7B were less and more frequent in Asians, respectively. Regarding the acetylator status, East Asians and Native Americans harboured the highest frequencies of the fast phenotype, followed by South Europeans. Central Asia, the Middle East, and West European populations were the major carriers of the slow acetylator status. The detailed panorama presented herein, expands the knowledge about the diversity patterns to genetic and acetylation levels. These data could help clarify the controversial findings between acetylator states and the susceptibility to diseases and reinforce the utility of *NAT*2 in precision medicine.

**Funding:** The authors received no specific funding for this work.

**Competing interests:** The authors have declared that no competing interests exist.

## Introduction

Arylamine *N*-acetyltransferase 2 (NAT2) is a phase II xenobiotic-metabolising enzyme with medical relevance, responsible for the biotransformation of several therapeutic drugs, environmental and diet compounds [1–5]. The *NAT*2 gene is strikingly diverse; 45 nucleotide variations have been reported hitherto, of which most are single nucleotide polymorphisms (SNPs) and two deletions (Δ859T and Δ3237A) found in South Indian and Japanese populations, respectively [6–9]. The combination of these variants affects the protein structure and the acetylation capacity, thereby producing at least three phenotypes: fast, intermediate, and slow [1]. Such acetylation states modify the efficient detoxification of exogen substances. Thus, the *NAT*2 genetic patterns could influence susceptibility to adverse drug effects and induce genetic damage such as DNA adduct formation [1,2,10]. Although genotype-phenotype associations have remained controversial, lifestyle and the acetylation phenotype have been associated with susceptibility to neoplasia, insulin resistance, and certain cardiometabolic traits [2–4]. On the other hand, absorption, distribution, metabolism, and excretion, cornerstones of the pharmacological effects, have shown relevant differences regarding the ancestral background [11]. *NAT*2 also exhibits allele, haplotype, and phenotype frequency variations across populations and ethnic groups. Demographic events, historical and cultural transmissions of the populations shape the genetic variation. Hence, some authors have pointed out that lifestyle, acetylation state, and genetic background, have delineated the current epidemiological transitions [12].

The 1000 Genomes Project database has portrayed the global diversity of the *NAT*2 polymorphisms; other studies have described its gene variability in specific populations (https://www.internationalgenome.org/). Nonetheless, several populations and ethnicities remain underrepresented. Furthermore, most studies have been limited to populational descriptive data, leaving gaps in the knowledge of the *NAT*2 genetic architecture that observational studies could make accurate.

Despite evidence about the *NAT*2 clinical relevance, the reconstruction of its worldwide diversity remains partial, requiring a comprehensive and detailed landscape. The present systematic review is state of the art, compiling the *NAT*2 genetic and acetylation patterns from 164 articles published from October 1992 to October 2020, representing 80 countries, 243 different populations and 101 ethnic minorities. The articles included descriptive studies and controls from observational studies expanding the diverse landscape of this phase II enzyme. We conducted the diversity analyses from 35,561 genotypes, 51,860 haplotypes and 70,484 phenotypes, providing one of the most complete and detailed panoramas to date. This review expands the knowledge about the diversity patterns of *NAT*2, applicable to drug therapies, pharmacogenetics, and susceptibility to diseases. Our data may even suggest the genetic patterns of unrepresented populations, where their close genetic ties with related populations could constitute the possible scenery of the harboured genetic architecture.

## Materials and methods

### Eligibility criteria

We conducted a systematic review following the Preferred Reporting Items for Systematic Reviews and Meta-analyses 2020 statement (PRISMA) [13]. Inclusion criteria were restricted to articles published in English and Spanish languages, conducted on human populations where at least two individuals were genotyped. These articles included both population genetic papers and observational studies with hospital- and/or population-based control groups evaluating the genetic contributions of NAT2 polymorphisms to allergy, asthma, hypersensitivity, and cancer. Data from observational studies were obtained from control groups to avoid any

skewed diversity related to a health condition. Several databases with an agreement with our institutions to acquire full-text papers (e.g., Embase, Lilacs, PubMed, and Scopus, among others) were used. In those studies where full-text access was denied, the corresponding author(s) was contacted on three occasions via e-mail, requesting the full-text; the article was removed from the analysis database if they failed to respond to our request.

Studies contravening the eligibility criteria in the primary research focus were excluded, as well as those whose genetic variant frequencies were < 1%. Commentaries, newsletters, reviews, overviews and overlapping publications were also removed from the analysis, along with systematic reviews. Articles lacking crucial information in their documentation, such as those lacking information about the number and reference sequence (rs) of SNPs used and those whose authors did not give access to its data, were also omitted.

## Information sources and search strategy

Articles indexed in Embase, Lilacs, PubMed, and Scopus databases published from October 1992 to October 2020 were included. The initial date was chosen because, from 1992 onwards, the number of articles, including the terms of the NAT2 gene and polymorphisms, increased exponentially. The search strategy included free-text terms such as "allergy", "asthma", "cancer", "hypersensitivity", "diversity", "ethnic group", "N-acetyltransferase 2" and "NAT2". These headings were combined with the terms "polymorphism, genetic", "polymorphism, single nucleotide", "genetic variation", and "DNA". Relevant articles selected from the reference list of the included items were searched manually to identify additional studies. All studies reviewed and included herein were from published data.

## Selection process

Two reviewers independently screened all studies retrieved from the research strategy using the title and/or abstract as eligibility criteria. These two reviewers further participated in the full-text revision of potentially eligible documents and assessed whether the articles met the inclusion criteria for their eligibility. Said reviewers independently carried out the data extraction and quality assessment of all the documents. Three more reviewers worked independently as arbiters to solve inconsistencies and screen disagreements. Disputes among these five reviewers were resolved through group discussion.

## Data collection process

**Data extraction.** Data extraction was conducted following the guidelines for observational studies in epidemiology. First author's name, publication year, country, ethnicity, geographic region, study design (i.e., case-control, cohort, cross-sectional and population-based), sample size, and gender were included in the data extraction [14,15]. Polymorphism was described as a gene variant with at least 1% frequency in the population; the location and "rs" of each single nucleotide polymorphism (SNPs) were also included.

Observed frequencies from alleles and genotypes were collected for each study (S1 Table). In those studies where allele frequency was not reported, it was set up using the haplotype frequencies reported; these data were underlined. Likewise, when the genotype frequencies were not described, these were constructed assuming Hardy-Weinberg equilibrium.

Concerning the haplotypes, observed frequencies, as reported by the authors, were included in the data extraction. Nonetheless, the statistical analyses were made only with those haplotypes representing the consensus nomenclature [https://nat.mbg.duth.gr] and assigned from at least six SNPs. These criteria were also used regarding the acetylator phenotypes frequencies (i.e., fast, intermedia and slow) reported by the authors. The fast phenotype was defined by the

presence of two fast acetylation haplotypes (i.e., *4, *11A, *12A, *12B, *12C, *13A, *18) in agreement with the consensus nomenclature [https://nat.mbg.duth.gr]. The slow phenotype was defined by two slow acetylation haplotypes (i.e., *5A to *5J, *6A to *6E, *7A and *7B, *10, *12D, *14A to *14G, *17 and *19). The intermedia phenotype was defined by the presence of one haplotype fast and one slow. In those studies, reporting very slow phenotypes, such data were added to slow phenotypes. Some authors reported the phenotypes frequencies using at least 6-SNPs and the *tag* SNP rs1495741; in such cases, the first option was solely considered; prior reports have suggested a similar panel to infer accurately the acetylator status [16]. Nonetheless, if the six-SNPs haplotypes included rs1801279, these data were excluded because this SNP was highly conserved amongst the worldwide populations. In those articles where only the *tag* SNP rs1495741 was reported, these data were used to obtain the three phenotype statuses where AA represented the slow phenotype, GG, the fast one and the heterozygous state, the intermedia phenotype. This *tag* has shown similar accuracy to those inferred with seven SNPs panel [16].

Those studies where the authors did not define the specific haplotype (i.e., *NAT2*5A*) and only reported the general haplotype (i.e., *NAT2*5*) were included in the database but excluded from the rest of the analyses to avoid a skewed panorama about the diversity. Likewise, in the studies where the authors only reported the haplotypes without the phenotype statuses, the frequency of these was obtained by adding the number of individuals with fast (to the fast phenotype) or slow haplotypes (to the slow phenotype) and dividing by in the total number of haplotypes reported. In this situation, only fast and slow phenotypes were reported without the intermedia phenotype.

**Quality evaluation.** The quality, internal validity, risk of bias and comparability were evaluated in each selected study using the Quality of Genetic association studies tool (Q-Genie) [17]. Q-Genie encloses the statements developed both STrengthening the REporting of Genetic Association studies (STREGA) as a means of strengthening the reporting of Genetic RIsk Prediction Studies (GRIPS) [14,15]. STREGA guidelines are built on the STrengthening of the Reporting of OBservational Studies in Epidemiology (STROBE) [18]. From these two lineaments (STREGA and GRIPS), Q-Genie evaluates the quality of genetic studies with eleven items, each one with seven numeric classification answers: one and two suggest poor quality, three and four suggest moderate quality and five to seven suggest high quality [17]. Three reviewers independently assessed the quality of all the articles selected; disagreements were resolved through group discussions with all five reviewers. Only those studies considered good quality were included: for diversity studies, the threshold score was $\geq 40$, whereas, for the studies with a control group, the score was $\geq 45$.

**Effect measures.** Given the characteristics of the study, measures of effect were not applied.

**Other statements.** This review was not registered, and the protocol was not prepared.

## Diversity patterns

Allele frequencies were collected for each study (S1 Table) and depicted in global maps obtained from the United States Geological Survey National Map Viewer [https://viewer.nationalmap.gov/viewer/], that is a public domain.

Although multi-ethnic studies were excluded from all analyses and comparisons, they were included in S1 Table.

## Statistical analysis

The frequency distribution of the fast and slow phenotypes of the different populations included in the present study was depicted by geographic regions (Africa; AFR, the Americas;

AMR, Asia; ASI, Europe; EUR, the Middle East; MEA, and Oceania; OCE) through violin plots. All continents were subdivided into regions according to the WorldAtlas webpage (http://worldatlas.com), and the frequency data was shown using box plots. Africa was separated into Central (CAf), East (EAf), Nort (NAf), South (SAf), and West (WAf), regions. The Americas were subdivided into Central (CAm), North (NAm) and South (SAm) regions. Its population diversity landscape was also separated into Afrodescendants (from the USA and Brazil), Asian Americans (from the USA), Native Americans, European-derived populations (whites from Canada and the USA represented as non-Hispanic whites, NHW), and Latinos. Asia was separated into Central (CAs), East (EAs), Southeast (SEAs), and South (SAs) regions. Europe was separated into East (EEu), North (NEu), South (SEu), and Western (WEu). Countries belonging to each region appeared in the footnote of each plot.

Violin and box plots were made with R software using GGplot2 [19]. The median differences among and within continents were performed using the chi-square test ($\chi^2$) with Med-Calc® Statistical Software v20.118 [20]. $P$-values $\leq 0.05$ were considered significant. Bar plots with proportions, area charts with the allele frequencies and doughnut charts were made using the Numbers app v12.2 (Apple Inc., 2022).

Haplotype diversity ($h$) and mean pairwise differences (MPD) were conducted only in the most frequent haplotypes to have a comparison panorama. These two calculous were made with Arlequin v3.5 using 1000 permutations [21]. MPD statistical differences among the different geographic regions were made using the Wilcoxon's test with MedCalc® Statistical Software v20.11 [20]. $P$-values $\leq 0.05$ were considered as significative.

**Comparison with other populations.** Data were compared with several populations bearing similar ancestral and geographic backgrounds from the 1000 Genomes Project database (1KGP; https://www.internationalgenome.org/). These comparisons were only made in Africa, the Americas, Asia, and Europe.

## Results

A total of 1090 publications, including 31 additional articles and 61 records identified by citation searching, were obtained from the first screening. Of these, 926 articles were excluded for various reasons (Fig 1). Two hundred and forty-three potential full-text articles were thoroughly assessed, of which 164 were included in the present study (S1 Table).

### The generalities of the studies included

The selected articles represented 80 countries, 243 different populations and 101 ethnic minorities. Of the total of studies, ~ 30% were from the Americas, followed by Asia (24%), Africa (21%), Europe (19%), the Middle East (5.263%), and Oceania (0.330%). From each geographic region, several sub-regions were analysed regarding the number of studies (Fig 2). The most studied area in Africa and Asia was the East (28% and 70%, respectively), whereas in the Americas and Europe was the South region (~ 54% and 35%, respectively).

The countries contributing to the significant number of populations studied in Africa were Cameroon, with ten populations, and Nigeria and Tanzania, with six populations (S1 Table). In the Americas were the USA and Brazil (26 and 18 populations, respectively), excluding the multi-ethnic studies reported in the USA, whereas in Asia were China and Japan with 18 and 12 populations, respectively. Germany (seven populations) and Spain (nine populations) were the European countries with the most population studied. However, the Russian Federation has contributed to 12 populations both in Europe and Asia regions. The Middle East has studied 15 populations.

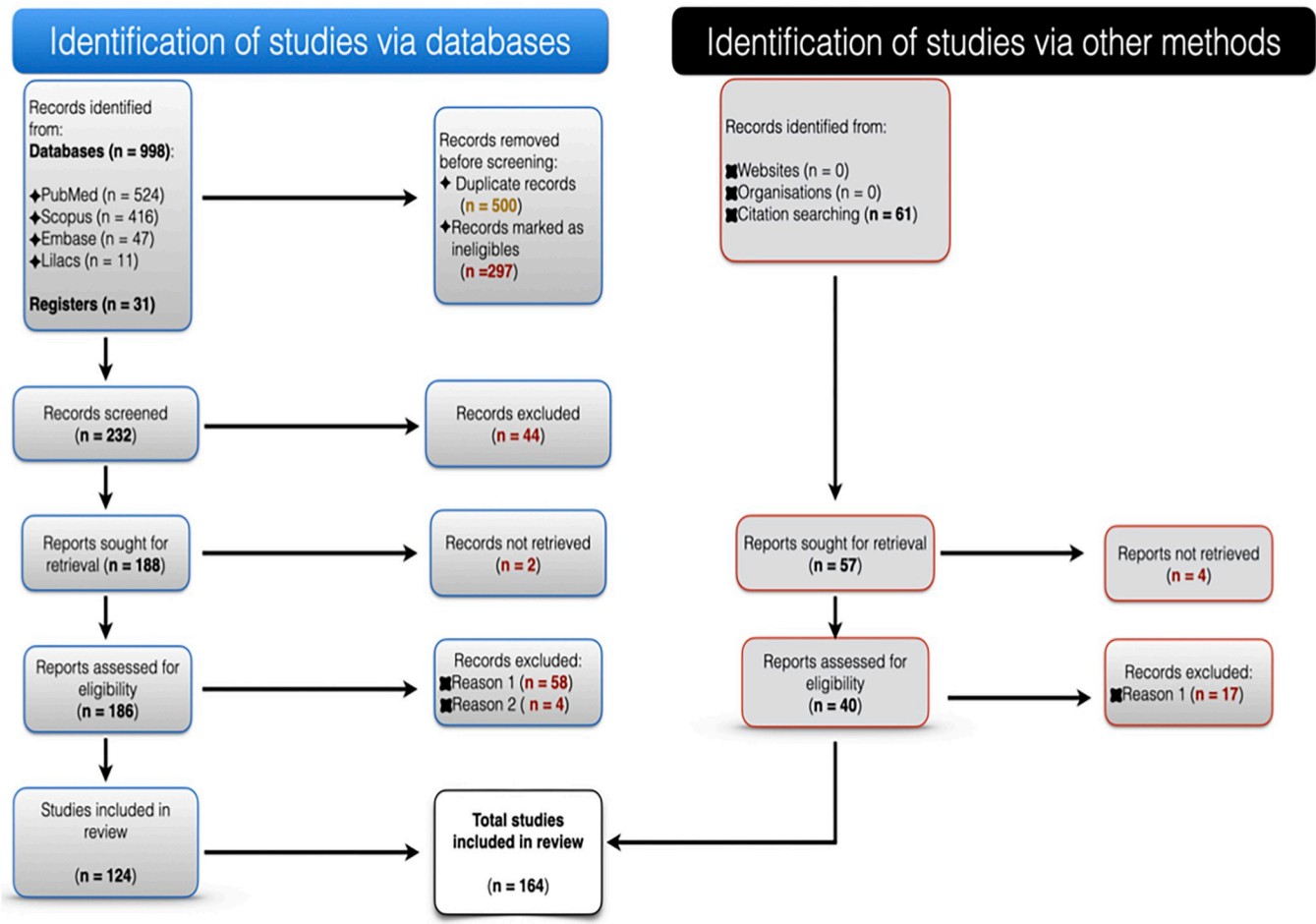

**Fig 1. Selection process used in the systematic review following the PRISMA 2020 statement. Note:** Reason 1, low quality. Reason 2, duplicated data.

Regarding the data source, 60% of the studies came from descriptive population studies; the European ones contributed to the most observational studies.

Based on ethnicity, 96% of the articles were from well-established geographic regions; the remaining studies involved multi-ethnic (more than three ethnicities) origins, which were excluded from all analyses. Present-day, European and European-derived populations have been the most studied.

## Allele and genotype diversity

The most studied polymorphisms within *NAT2* were rs1801279, rs1041983, rs1801280, rs1799929, rs1799930, rs1208, rs1799931, and rs149574 (S1–S5 Figs). Of these, rs1801279 and rs1799931 depicted a conserved distribution of ancestral alleles being the most prominent. In the case of rs1801279, the African populations and the United Arab Emirates presented the major frequencies of the derivative allele. By contrast, Asians and Latinos showed the highest frequencies of the derivative allele regarding the SNP rs179931. Of note is the high frequency of this allele in Swedish (0.364) and Emiratis (0.244). Another SNP with similar distribution worldwide was rs1041983, whose ancestral allele frequencies presented a range from 0.591 (in Asians) to 0.710 (in Europeans). SNPs such as rs1801280, rs1799929, and rs1208 exhibited

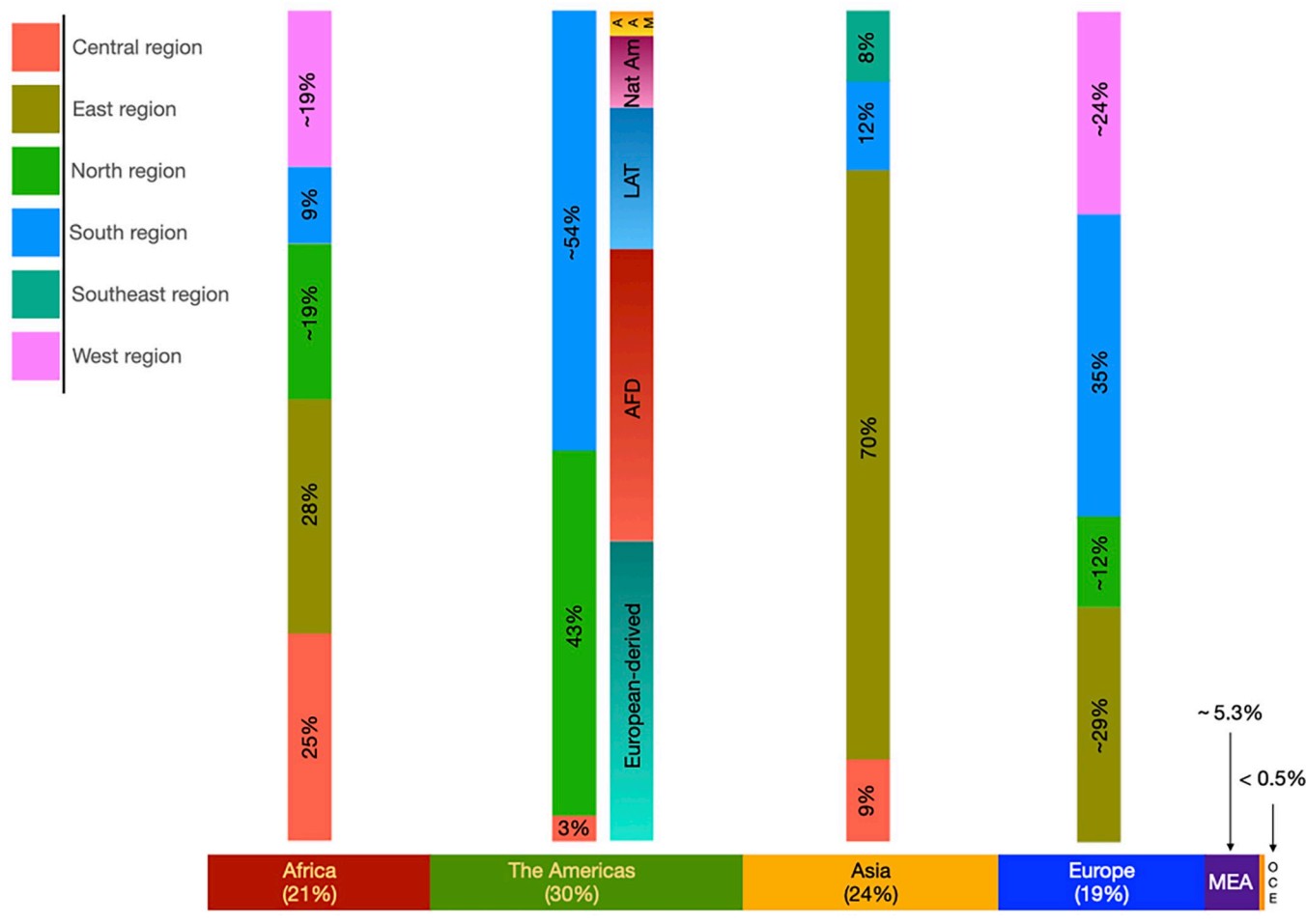

**Fig 2. Percentage of articles by region included in the systematic review considering geographic regions and subregions. Note:** AAM, Asian Americans; AFD, Afro-descendants; LAT, Latinos; MEA, Middle East, Nat Am, Native Americans; OCE, Oceania.

several distribution patterns with remarkably high frequencies of the ancestral allele in the Asian populations.

Regarding the derivative alleles of these three SNPs, they presented the highest frequencies within European populations. About rs1799930 polymorphism, the highest frequency of the ancestral allele was shown in Latinos (range: 0.775 in Brazilians to 0.999 in Ecuadorians) and Papuans, this last population with a low portrayal (n = 2). The derivative allele was uniformly distributed worldwide (except in Latinos), although Swedish exhibited a remarkable frequency (0.663).

Concerning the *tag* SNP rs1495741 (Fig 3), it has been the least studied with high frequencies of the derivative allele representing the slow phenotype, which was more frequent in South Asia (*f* = 0.779) and Europe (*f* = 0.756). Worthy of note are the distributions of the derivative allele in the Mali population and the opposite pattern in Brazil.

## Haplotype diversity by geographic region

The present systematic review depicted the distribution of 97 different haplotypes (S1 and S2 Tables); 68 were determined using at least 6-SNPs. Such 6-SNPs haplotypes were obtained from 19,301 individuals (38,601 haplotypes). Of these, 34 singletons were found. Overall, the

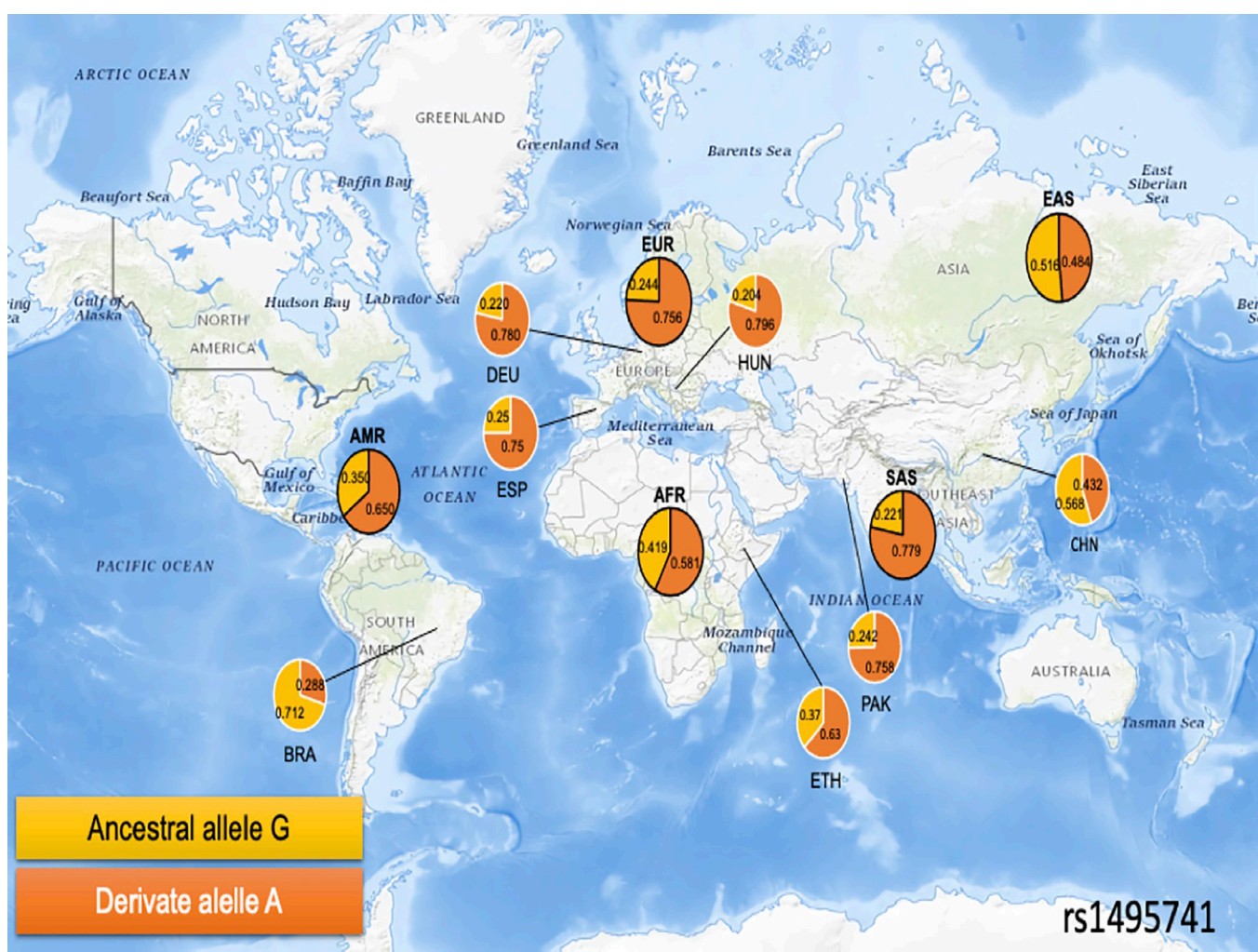

**Fig 3. Frequency of the ancestral and derivative allele of rs1495741. Note:** AFR, Africa; AMR, The Americas; BRA, Brazil; CHN, China; DEU, Germany; EAS, East Asia; ESP, Spain; ETH, Ethiopia; EUR, Europe; HUN, Hungary; MLI, Mali; PAK, Pakistan; SAS, South Asia. The map was obtained from the United States Geological Survey National Map Viewer [https://viewer.nationalmap.gov/viewer/], that is a public domain.

haplotype *NAT*2*4 (wild type) was the most common globally, followed by *5B, *6A, and *7B (S6 Fig). Other haplotypes with critical frequencies were *12A (in African, European and the Americas populations), *5A, *5C, *7A (Asian and the Americas populations) and *14B (in Africa and the Middle East).

Regarding the different haplotypes found by region, Africa presented 41 haplotypes, of which twenty have been described, as yet, only in this region (singletons). Such singletons have been characterised, mainly within *6 and *12 haplotype clusters. The Middle East was represented by 32 haplotypes and nine singletons within the *5 haplotype cluster. The Americas exhibited 31 haplotypes and six singletons within *6 and *7 haplotype clusters, whereas Europe presented 26 haplotypes and three singletons. Asia was the least diverse region, with seventeen haplotypes and one singleton. Oceania was only represented by two haplotypes from two individuals. Such patterns may present bias because they depend on the resolution power of each study, the sample size and the dates on which they were made; the new technologies have the advantage of the resolution within haplotypes. Hence, the haplotype diversity and mean pairwise differences were conducted with the data from the eight most represented haplotypes

**Table 1. Diversity patterns for *NAT*2 haplotypes by continental regions.**

| Region | N | h | Haplotype diversity | MPD |
|---|---|---|---|---|
| Africa | 1,955 | 15 | 0.728 ± 0.007 | 2.508 ± 1.352 |
| America | 5,362 | 20 | 0.689 ± 0.004 | 2.648 ± 1.414 |
| Asia | 3,206 | 13 | 0.648 ± 0.006 | 2.002 ± 1.129 |
| Europe | 13,176 | 21 | 0.583 ± 0.003 | 2.444 ± 1.324 |
| Middle East | 2,957 | 16 | 0.715 ± 0.005 | 2.736 ± 1.452 |

**Note:** *N*, number of samples; *h*, Number of different haplotypes found; MPD, mean pairwise differences.

excepting *NAT2*\*4 (i.e., \*12A, \*5A, \*5B, \*5C, \*6A, \*7A, \*7B, and \*14B). These results showed comparable patterns to those described as a whole without any significant difference amongst the different regions (Table 1). In Africa, the western and central countries contributed to the greatest diversity, principally in those haplotypes within \*6 and \*12. The countries within the southern region were the least diverse but also the minor studied. In Europe, the most diverse region was the southern one, followed by the Easter and Western.

No significant values ($p$-values $\geq$ 0.05) were found among MPD by geographic regions by the Wilcoxon's test.

### Acetylation capacities

**Slow phenotype.** The substantial charge of the slow phenotype worldwide was a consequence of the high frequency of the slow haplotypes. This status was more frequent in the Middle East, where most data were distributed around the median (0.782) with remarkable frequencies in the Ashkenazi Jews, Emiratis and Pakistanis (S1 Table). Significant differences ($p \leq 0.0001$) were found between MEA and Asia (~1.4 times lower) and the Americas (~1.3 times lower) when comparing the median values (Fig 4). Africa and Europe also presented high frequencies of this phenotype (median values: 0.758 and 0.751, respectively, without significant differences). About Africa, the north region presented the highest median value (0.823), showing marked differences with CAf, SAf, and WAf ($p \leq 0.001$) (S7 Fig). The prominent frequencies were presented in Cameroon (CAf) within the Fulani ethnicity and Tanzania (EAf) in Burunges, Hazdas and Maasais. Of note, the contrasting frequencies are even within the same country (i.e., Cameroon). Conversely, SAf presented the lowest median value (0.391) with significant differences ($p \leq 0.0001$) with all regions.

Europe regions presented similar slow phenotype distributions with median values in a rank of 0.736 (SEu) to 0.795 (NEu). Thus, any significant difference ($p \geq 0.05$) among the different regions was identified (S8 Fig). The highest frequencies were seen in NEu (Sweden) and WEu (France), whereas the lowest ones were found in SEu (Serbia) and France. The Americas (0.612) and Asia (0.565) presented the lowest median values (S9 and S10 Figs). The lowest frequencies were observed in Japan (EAs) and within the Native Americans. Inside the Americas, the lowest median values (0.147) were seen in CAm, which was ~ 4.5 and 3.8 times lower in comparison with NAm and SAm, respectively ($p \leq 0.0001$); this interpretation should be taken with caution. The patterns of this phenotype were dissimilar depending on the ancestral background. Latinos presented the most significant frequencies of the slow phenotype: Brazil (0.480) and Mexico (0.560). High frequencies were also observed within the Afro-descendants from Brazil (0.290) and the USA (0.260) and in the European-derived populations from Canada and the USA (0.310) (Fig 5).

Regarding the Asian region, the highest values of the slow phenotype were presented in SAs (0.839), represented by two Indian studies. SEAs (0.640), CAs (0.615) also presented similar

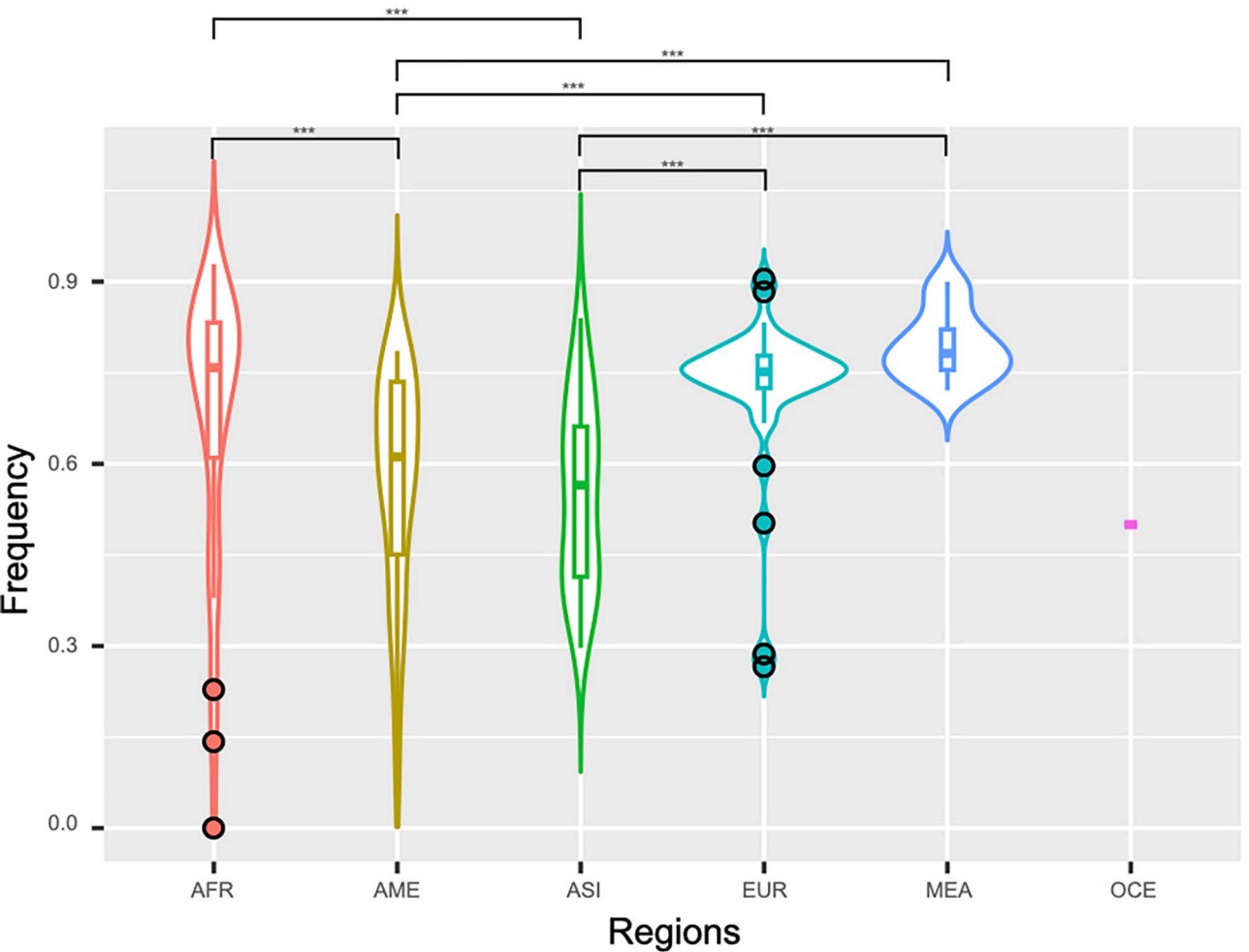

**Fig 4. Violin plots of the slow phenotype distribution by geographic region. Note:** AFR, Africa; AME, The Americas; ASI, Asia; EUR, Europe; MEA, Middle East, OCE, Oceania.

frequencies (S10 Fig). By contrast, EAs (0.429) exhibited the lowest values, particularly within Japanese populations, except in the Eskimos and Yakuts from Siberia. Yet, the pattern of this region should not be generalised, given the paucity number of studies.

**Fast phenotype.** Regarding the fast phenotype, it was more frequent in Asia (median = 0.435) and the Americas (median = 0.388). Thus, significant differences ($p \leq 0.0001$) were found to compare with Africa (median = 0.242), Europe (median = 0.249) and MEA (median = 0.218), where it was almost twice less frequent (Fig 6). Some African populations, such as Baka and Bakola Pygmies from Cameroon, Biaka Pygmies (Central African Republic), and San (Namibia), also presented high frequencies of this phenotype (S1 Table). A similar pattern was found in Serbia and France within the European region.

Inside Asia, the Eastern region presented the higher median frequencies (0.571); Chinese, Han Chinese, Koreans and Western Siberians exhibited the highest values. Thus, significant differences ($p \leq 0.0001$) were found to compare it with the other regions (CAs = 0.385; SEAs = 0.360; and SAs = 0.161) (S11 Fig). About the Americas, overall, the Native American populations exhibited the major frequencies; Emberas and Ngawbes from Panama presented

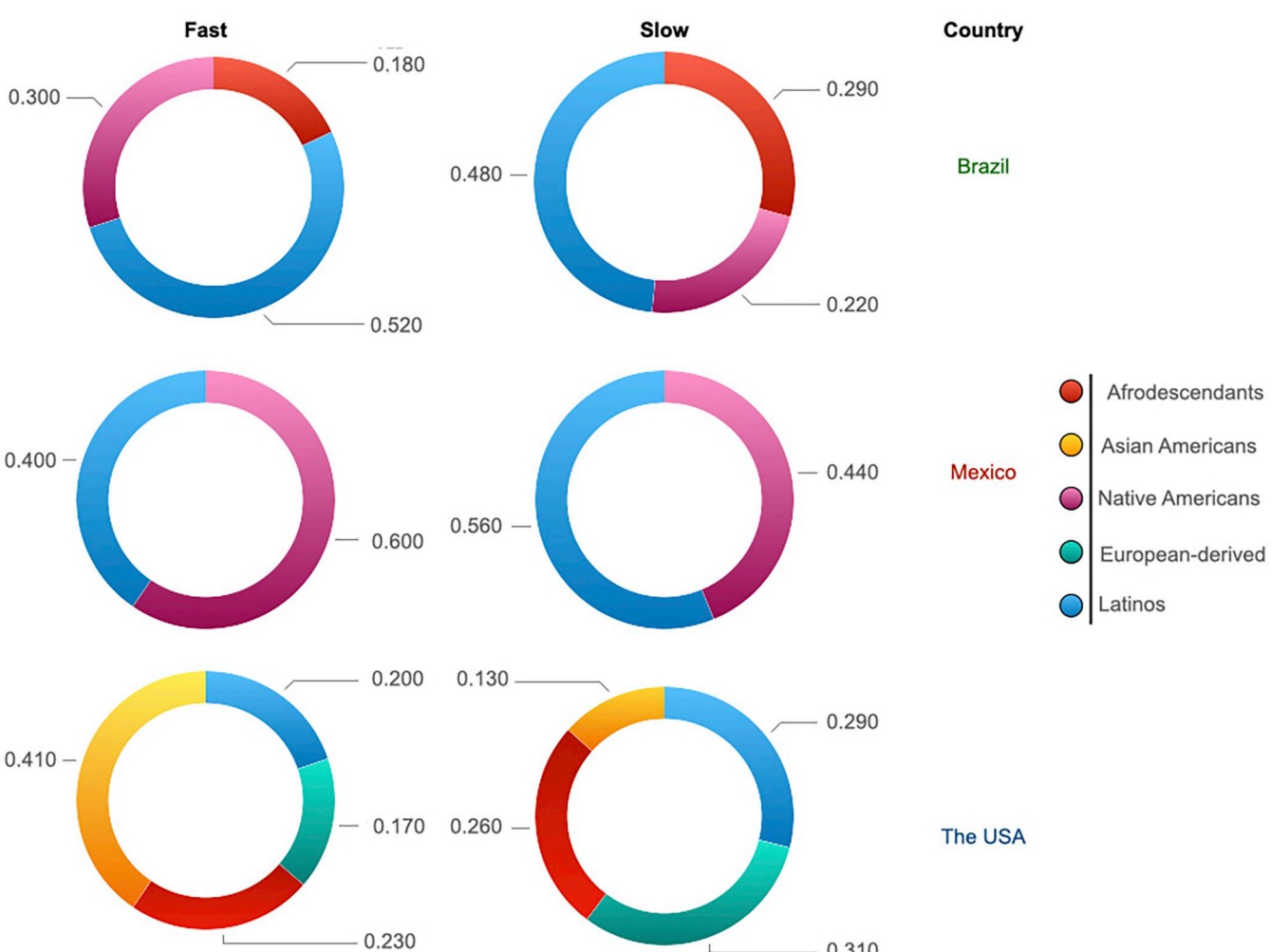

**Fig 5. Doughnut charts representing the fast and slow phenotypes distributions by ethnicity in Brazil, Mexico and the United States of the America.**

the highest frequencies (0.853 and 0.924, respectively) (S12 Fig). Latinos also presented high values of this phenotype (Fig 5).

About Africa, the highest frequencies of the fast phenotype were found within SAf populations (median = 0.609), followed by the Central region where Bakola Pygmies (Cameroon) presented the greatest values, accompanied by Namibia (SAf, 0.857) and the Wolaitas from Ethiopia (EAf, 0.621) (S13 Fig). Significant values were found amongst the different regions (range $p \leq 0.05$—$p \leq 0.0001$). Among the different countries belonging to the Middle East, the highest frequencies were presented in Jordan (0.279), and Druze from Israel (0.273); the Ashkenazi Jews (0.100) and the Emiratis (0.119) presented the lowest frequencies (S1 Table).

Europe depicted similar median distributions among the regions (range:0.205 in Neu to 0.264 in SEu) without any significative differences (S14 Fig). Of note is the high frequencies of the fast phenotypes in Serbia (0.730) and France (0.711).

**Intermedia phenotype.** Although this phenotype has not been fully reported, those studies that included it have illustrated high frequencies in East and South African populations as well as African Americans. By contrast, the lowest values were found among Europeans, its descendant populations, and in the Middle East (Fig 7).

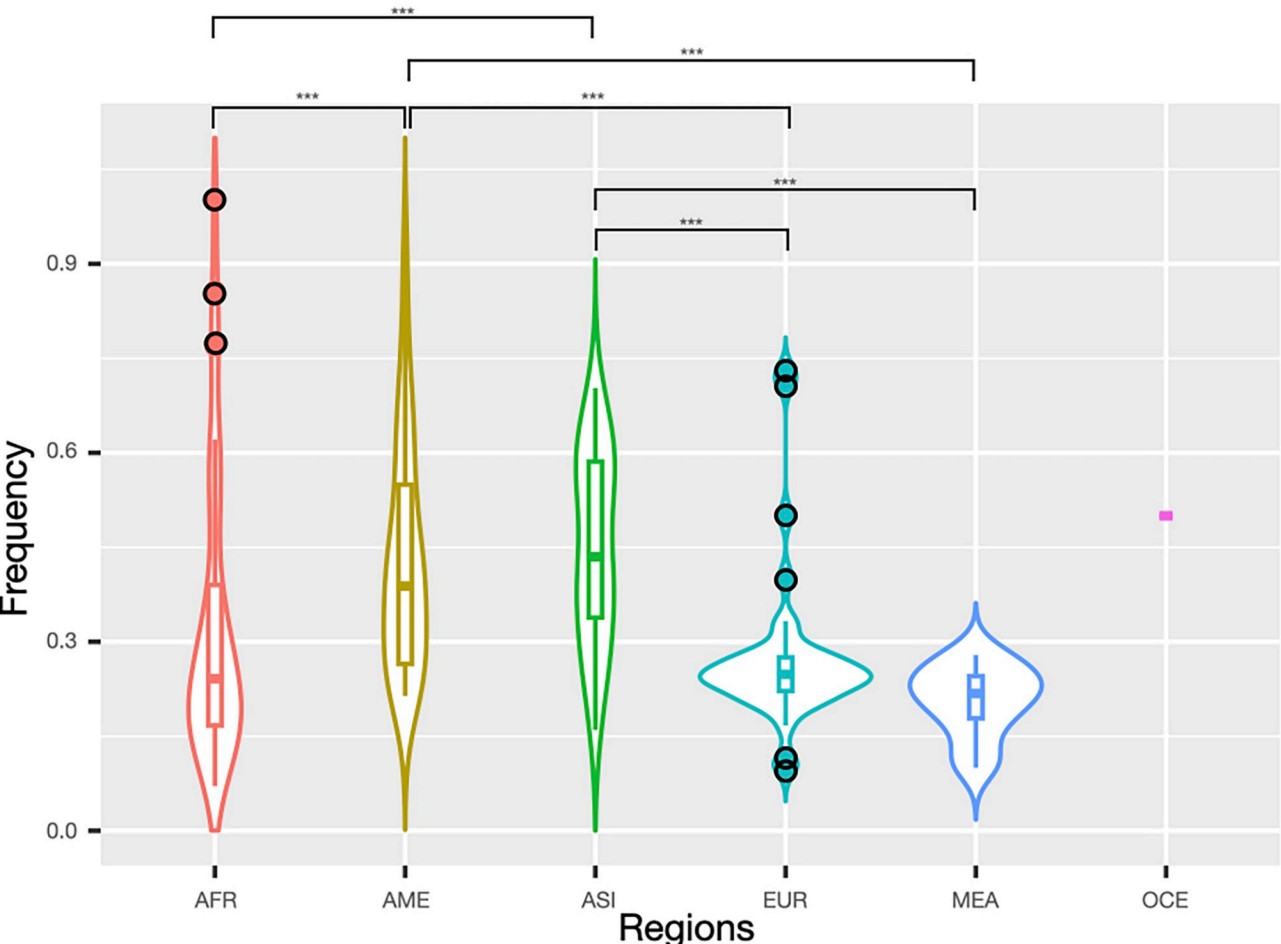

**Fig 6. Violin plots of the fast phenotype distribution by geographic region. Note:** AFR, Africa; AME, The Americas; ASI, Asia; EUR, Europe; MEA, Middle East, OCE, Oceania.

### Comparative with 1000 genomes project data

Overall, the allele frequency distributions of the eight most studied *NAT2* polymorphisms were congruent with those reported in 1KGP. However, the panorama of populations analysed herein portrayed a more accurate frequency distribution and included more populations and individuals. Our contribution to the scenery of genetic diversity in Africa comprised 22 populations. The Americas had several populations (Natives, Afro-descendants, Asian Americans, and European-derived populations) from Canada, the USA, Mexico, Nicaragua, Panama, Colombia, Argentina, Paraguay, Brazil, Peru and Ecuador. Inside Asia, Central and Southeast regions were included herein, enlarging the diverse representation landscape of East Asian and South populations. About Europe, our study represented populations from the East, West, North and South of the continent.

Of note is the behaviour of the distribution of rs1495741 in the Americas and African populations, which presented different patterns (e.g., Brazil and Mali, respectively) to those described by 1KGP. In addition, the Middle East populations were represented for the first time.

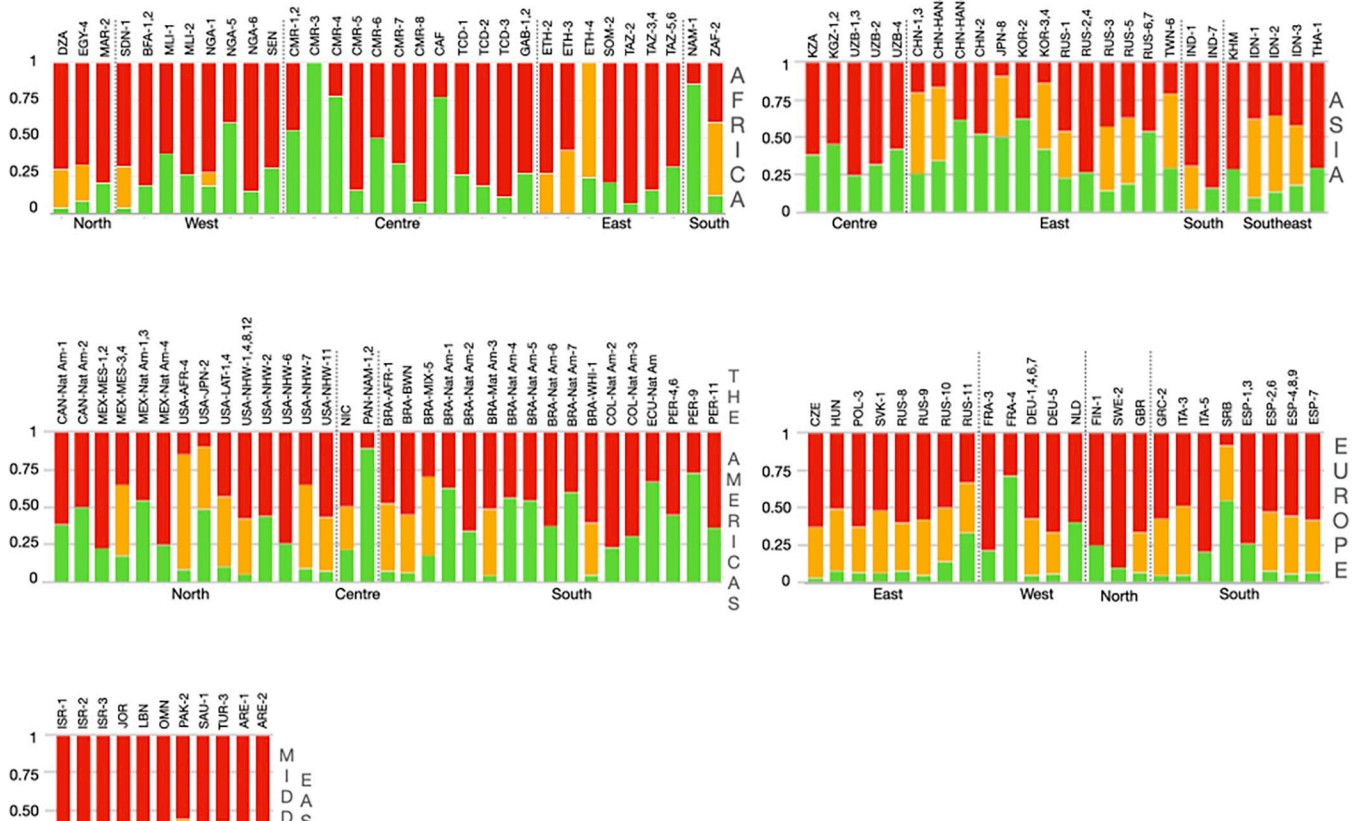

**Fig 7. Bar plots with the distributions of the slow, fast and intermedia phenotypes by geographic region. Note:** ARE, United Arab Emirates; BFA, Burkina Faso; BRA, Brazil; CAF, Central African Republic; CAN, Canada; CHN, China; CMR, Cameroon; COL, Colombia; Czech Republic; DZA, Algeria; DEU, Germany; EGY, Egypt; ESP, Spain; ETH, Ethiopia; FIN, Finland; FRA, France; GAB, Gabon; GBR; United Kingdom; GRC, Greece; HUN, Hungary; IDN, Indonesia; IND, India; ISR, Israel; ITA, Italy; JOR, Jordan; JPN, Japan; KGZ, Kyrgyzstan; KHM, Cambodia; KOR, Korea; KZA, Kazakhstan; LBN, Lebanon; MAR, Morocco; MEX, Mexico; MLI, Mali; NAM, Namibia; NGA, Nigeria; NIC, Nicaragua; NLD, Netherlands; OMN, Oman; PAK, Pakistan; PAN, Panama; PER, Peru; POL, Poland; RUS, Russian Federation; SAU, Saudi Arabia; SEN, Senegal; SDN, Sudan; SOM, Somalia; SRB, Serbia; SVK, Slovakia; SWE, Sweden; TCD, Chad; THA, Thailand; TUR, Turkey; TWN, Taiwan; TZA, Tanzania; USA, The United State of America; UZB, Uzbekistan; ZAF, South Africa. LAT, Latinos; Nat Am, Native Americans NHW, mom-Hispanic Whites representing the European-derived populations from Canada and the United States of America.

## Discussion

*NAT*2 polymorphisms have shown variations in the allelic distributions across populations and at inter-ethnic and inter-individual levels. Although, several studies have described the gene variability of *NAT*2, these have been limited to populational descriptive data, leaving gaps in the knowledge of its genetic architecture. The present systematic review explored, in detail, the global *NAT*2 diversity patterns from 304 populations, including 164 articles from populational-descriptive and observational studies.

### Particularities of the studies included

Akin to other documents related to several pathologies and pharmacogenetic studies, Europeans (including European-derived populations from Canada and the USA) have been the most characterised, mainly to avoid spurious results given its genetic homogeneity [22–26]. Such

homogeneity was demonstrated in the distribution of both the slow and fast phenotypes in the different regions of Europe without significant differences among them (S8 and S14 Figs).

East Asia, particularly the Han Chinese populations was the second most studied region. This ethnic group has also been considered homogeneous, given its age (traced back to the Neolithic) and the gene-flow with surrounding populations [27,28]. The remarkable number of studies in East Asia could be associated with the earliest stages of sedentism and plant cultivation found in northern China, the second oldest domestication centre in Eurasia [29]. In this region, several agricultural systems (e.g., millets and rice) emerged independently and gradually increased in the Hexi Corridor and the Yellow River Basin [29,30]. Several human groups interacted during this period, and with the human expansions, the cultures spread southward and northwards (almost simultaneously) as well as central China [29,31,32]. The genetic background of the Han Chinese population has been associated with different subsistence strategies such as hunter-gatherers (Mongolia and Amur River Basin), farmers (from Yellow River Basin) and pastoralists (from western Mongolia), all around 3000 years Before Christ (BC) [33]. Thus, East Asians may derive from different mixture proportions, which makes their study remarkably interesting regarding the acetylator statuses, but also particularly complex. Han Chinese populations depicted a genetic cline where farmers from the Upper and Middle Yellow River share a gene pool with the north Han group [33]. The Yellow River Valley connected to China and Southeast Asia, and in turn, Han Chinese also shared a gene pool with Southeast Asians, probably, through southern Chinese agriculturalists [33]. These particularities could explain several phenotype patterns described in Han populations.

Other East Asian populations (e.g., Japan, Korea and Taiwan) and its counterparts in Western countries (i.e., Asian-Americans) were also represented. Nevertheless, the findings in these last populations might lead to flagrant under- or overestimation of their diversity, given the remarkable inbreeding rates regarding the outbred populations from which they arise [34,35]. Similar findings have been described in the equivalent of South Asian populations (i.e., India and Pakistan) [35,36]. In turn, the reported data regarding Asian Americans should be interpreted with prudence because their patterns could be closely related to the genetic architecture of each population, limiting its applicability.

Historically ethnicities such as Africans, Latinos and Native Americans' populations have been understudied. Nonetheless, our findings depicted an increase in the number of studies involving them in recent years [37–39]. Particularly, the subsistence mode and the diversity in climatic zones and biomes have been proposed as keystones in the evolution of *NAT2*, exerting a positive selection [9]. Thus, studies on African ethnic groups have been interested in elucidating the adaptation signals related to diet and lifestyle practices [40–43]. African populations are ethnically and genetically diverse, forging a cornerstone for answering such questions [42]. Thus, it was not wholly surprising that the most extraordinary *NAT*2 diversity has been found in Africa.

Latinos are a cradle of diversity; their genetic background was shaped by the fusion amongst European and East Asian migratory waves peopling the New World and African populations [44,45]. Thus, the Americas are a melting pot of diversity that emerged 500 years ago (ya). Both European and African migrants came from several regions. The European migrants that colonised the Americas came from England, France, Portugal and Spain. Regarding the Iberian peninsula's geographic position, it favoured the trade interchange with circum-Mediterranean cultures (i.e., Greeks, Phoenicians and Carthaginians) besides the colonisation by the Romans and the Muslim domination [46]. English, French and the Netherlands migrations, as pirates and corsair, also invaded the Caribbean region, which also concentrated the enslaved Africans from Angola, Congo, Gambia, Ghana, Guineas, Mozambique, and Senegal, amongst others [46–49].

Furthermore, the Jewish and Muslim Diaspora in Latin America as *Conversos* lead to genetic diversification [46,50]. Native Americans traced their origins back to a host of regions from Eurasia, explaining their numerous diversities in languages and lineages [49,51]. A remarkable intrapopulation diversity has been observed in the Native Americans, supporting the results found in this review [37,52]. Likewise, the genetic wealth of contemporary Native Americans, in conjunction with the several admixture degrees in their non-Native American populations, makes the Americas an excellent candidate for pharmacogenetic studies [39,49,53,54].

Hence, Africans and Americans were the most diverse populations regarding *NAT*2 diversity, reinforcing the relevance of including ethnic minorities in studies on diversity. The presence of new variants in these two ethnic backgrounds depicted their diversities and possibly a recent growth within their demographic history. Nonetheless, such variants could mirror the ability of modern technologies (e.g., sequence of the whole gene and array-based genotyping platforms) compared to genotyping with a limited number of SNPs. The findings from prior studies do not rule out the diversity of the populations, which could be skewed by the number of SNPs selected (i.e., not-informative markers). In turn, the number of singletons described and the diversity within haplotypes should be taken with caution.

Concerning those similar patterns described in Native Americans and East Asians, there seem to be related to the Americas peopling, and the bottlenecks and genetic drive underwent for the first settler populations [49,55]. Nonetheless, the Asian and Native American populations included herein did not represent the full diversity because most belonged to ethnic and religious groups besides the small sample sizes described. Similar arguments also might explain the diverse patterns found in Asia, where the least number of haplotypes were described. As mentioned earlier, the diversity depends on sample sizes and the genetic drift effect in these small and endogamic populations, the distinct ethnic backgrounds and the resolution power of the technologies employed [9].

Despite the reduced number of studies, the remarkable diversity of the Middle East was remarkable. Akin to Africa, this feature could be related to lifestyle practices. The transition from hunter-gatherers to an agricultural lifestyle has been associated with the domestication of wild cereals and plants [56]. Wheat, barley, among other cereals were domesticated in the Fertile Crescent during the Neolithic era (circa ~8,000–10,000 ya) [57–60]. This region spans the current countries of Iraq, Syria, Lebanon, Israel, Palestine, Jordan, Kuwait (northern region), Iran (western region), and Turkey (southern region). Early evidence of cultivation and domestication have been reported in Syria (11,150–10,450 BC), Jordan valley (9,700–8800 BC), South-eastern Turkey, the Upper Euphrates valley (Abu Hureyra, Syria), and Jericho (Israel) [59,60]. Besides, the Levant was one of the earliest regions where agriculture and animal domestication emerged [61]. Archaeological and genetic studies have reinforced that the north of Iraq (8,000–11,000 years Before Present) was the core of sheep's initial domestication [62]. Both plant cultivation and animal domestication are cornerstones of human societies' modification because these were learned from one region to another, favouring the genetic exchange, which could explain the diversity patterns [59].

## SNP diversity patterns

As in other studies, SNPs such as rs1801279 and rs1799931 depicted particular distributions, especially in Africa and the Americas, respectively [9]. The other SNPs' distribution patterns were similar to those previously reported by other research groups [9]. Yet, the peculiar pattern of rs1495741 distribution in Brazilian populations might be a reflection of the demographic events in the peopling of this young population (~ 500 years) and the subsequent bottleneck events [46].

## Slow acetylators explanations

*NAT* genes (*NAT*1 and *NAT*2), as others (i.e., *FADS*1), have been targets for natural selection [63]. Several studies have explained the prevalence of the slow acetylator status through selective pressures exerted by environment and lifestyle. Briefly, the transition from the fast acetylators to slow ones has been associated with the emergence of agriculture and pastoralism, replacing the hunting-gathering subsistence mode [40,63,64]. The shift in lifestyles involved the introduction of new foods to the diet with different nutrients and fats, as well as exposures to new pathogen [63]. These changes, in turn, could be implicated in the participation of metabolic pathway genes and those polymorphisms related to slow haplotypes, conferring certain advantages [63,65,66]. Finally, these genetic variants were fixed in the populations, increasing their frequency to favour adaptation. Hence, the slow acetylator phenotype has been more frequent within food-producing populations from Central and Southern Asia, North and Central Africa, Europe, and the Middle East [65–67]. By contrast, in hunter-gatherer populations, the rapid acetylator phenotype has been the most frequent [41,65]. The heterogeneous phenotypes in Africa and the Americas, which could be explained by their intrinsic diversity and the remarkable differences among and within populations [9].

## Fast acetylators explanations

Although slow acetylator status is the most frequent worldwide, fast acetylators have remained mainly in EAS and Native American populations. These results support the findings of former studies [41,68]. The hunter-gatherer lifestyles in ancient populations have been documented in Eastern Asians [63]. Regarding the Native Americans, it is likely that they have maintained the lifestyles of their ancestors. Cultural diffusion (i.e., subsistence practices) also shapes gene frequency patterns [69]. Studies have reinforced the ancestral connection between Native Americans and the East Asian populations [49,70–72]. Given the fossil evidence described in this region (i.e., bison, horse, and mammoth, among others), ancient North Americans' diet has been associated with the hunter-gatherer subsistence mode [73–75]. Other processes could explain the genetic architecture. Demographic events such as bottlenecks sustained prior to or during the colonisation of the Americas could also involve the frequency of fast acetylators distribution in these populations [76]. Instead, the effect of genetic drift is most substantial in small populations. At the same time, the number of migrants during the peopling of the Americas is controversial; the effective population size could have been small [49,77]. These migratory waves could be carriers of heritable traits "fixed" amongst populations or sub-populations under selective pressures, transmitting the "modified" haplotypes to other geographic regions [78]. Notably, the folate-rich diet and green leafy vegetables have also been associated with this phenotype [41,66,79,80]. Fish and soy, both folate-rich sources, are key ingredients of East Asian cuisine [81–83].

These adaptations could also have arisen independently in other geographic regions. Thus, one possibility that explains the subtle difference patterns in the frequency of the fast phenotype in Africa could be that the north region has mainly been occupied by hunter-gatherer populations of at least 5,000 ya [84]. Amongst these, the ethnic groups from Cameroon, Gabon, and Namibia depicted a significant frequency of fast acetylators. A comparable argument might explain Europe's highest proportions of the fast phenotype in Europe. Hunter-gatherer ancient populations have been established in Central, North, and South European regions [63]. By contrast, the East and West European regions have been associated chiefly with agricultural practices and, in turn, with carriers of slow phenotypes [63]. Nonetheless, the frequency patterns in this geographic region did not necessarily correspond to the agriculture diffusion, suggesting that the different gene flow levels could have influenced the frequency

distributions in Eurasian populations [85]. As mentioned before, demographic events play a critical role in gene frequency variations. The out-of-Africa and the Arabian aridification by climate change are two bottlenecks that could have impacted the diversity patterns of the Middle East and Eurasian populations [86,87].

Because not all natural xenobiotics were related to Nat2, other xenobiotic biotransformation genes could have been affected by the selective pressures exerted by environment and lifestyle [88,89]. Such selection effect could bring about behavioural adjustments in physiological and biochemical pathways as well as in the gut microbiome [63,90–92]. Complex biological pathways regulate metabolisms; in turn, hundreds of genes are likely involved in this physiological process. Nat2 is expressed in the intestines and liver; thus, possible coevolution would entail. Consequently, it is unlikely that the adaptation proceeding acted on single genes but rather, was a polygenic selection process that could also shape the *NAT*2 frequency patterns. Genes related to diet and metabolisms have been persistent in the models of polygenic selection [63,75].

A similar argument might explain the dissimilarities between the frequencies of *NAT*2*5B in worldwide populations regarding some areas of Asia, where its haplotype is less frequent, replicating the findings of prior studies [41]. On the one hand, the rs1801280-C allele, encoding for the altered slow phenotype, is more frequent in Central and Western Eurasians (range 0.287 to 0.500) than in East Asians (range 0.037 to 0.269). Such differences could result from a selection process, given that the rs1801280-C allele has increased its frequency significantly in Eurasians [65,89]. Again, the emergence of agriculture could be the selective pressure to the shift from the ancestral state rs1801280-T to the derivative one [65]. By contrast, the low frequency of the *NAT*2*5 haplotype in East Asians could be related to its liking for the hunter-gatherer lifestyle and other aspects mentioned before [63,93–96]. The *NAT*2*5 haplotype has shown an association with *NAT*1*4 in western and central Eurasians by 80% [65]. This association is twice and four times more than those found in Eastern Eurasians and sub-Saharan populations, respectively [65,89]. The effective metabolisms of environmental xenobiotics should require the collective action of phase I and II enzymes. Since these two genes are located on the same chromosome coevolution, it should not be unlikely [89]. Hence, the *NAT*2*5B selective advantage could affect the evolution of *NAT*1 as other genes [65]. Other studies have suggested that *NAT*1 and *NAT*2 could evolve under distinct selective regimes. These two genes have a physical distance fairly close to 200 kilobases, exhibiting linkage equilibrium among them [9].

In addition to 341 C > T, another three sets of polymorphisms (191 G>A, 590 G>A, and 857 G>A) are encoded for the slow acetylator state. Of these, 857G>A have depicted more frequency in Asians than Europeans [89]. The hunter-gatherer subsistence mode could also explain these differences, which acquires food from their surrounding environment. However, it did not discard the effect of demographic events. Contemporary populations maintaining this subsistence mode have shown a correlation between population density and local primary production [97]. While contemporary populations are not analogues to the ancient ones, the climate conditions to which ancient populations were exposed could use up the aliments from the environment with the subsequent bottleneck. Similar to the out-of-Africa model, the famine could have favoured migrations like those peopling the Americas.

Nevertheless, in addition to the diet, the diversity patterns could also reflect the environmental xenobiotic insults, the epigenetic regulation, the history and specific pressures of the populations, and climate features, among others [9].

Among the strengths of the present systematic review is the detailed landscape depicting the diversity of *NAT*2 from 35,561 genotypes, 51,860 haplotypes, and 70,484 phenotypes. These data portrayed the eight most studied SNPs, and for the first time, the *NAT*2 diversity of the Middle East populations, which has not been reported in any former studies. Likewise, the present diversity panorama discriminates between the most prominent ancestries in the

Americas: Latinos, Native Americans, and non-Hispanic whites. These features reflect the diversity among populations and individuals and could be a cornerstone for having a possible scenario regarding other ethnicities [11]. Diversity within *NAT*2 has been related to the developing drug side effects such as hepatotoxicity, peripheral neuropathy, lupus, and susceptibility to some kinds of cancer [1,98,99]. It is also necessary to highlight some of the study's shortcomings, including that not all studies included information on the diversity patterns of the eight polymorphisms and data from the specific haplotypes.

## Conclusion

The global diversity that occurred in ancestry and demographic events begs an understanding of the variation within genes of tremendous importance, such as *NAT*2. The present study provided the most up-to-date overview of the *NAT*2 diversity to allele, genotype, haplotype, and acetylator status with implications in pharmacogenetics and certain complex disease susceptibility. The study of this set of approaches could further illuminate its value and usefulness in personalised and precision medicine. Nonetheless, further studies are needed to unravel such diversity in ethnic minorities besides correlating the worldwide population diversity with pharmacodynamics and pharmacokinetics strategies.

## Supporting information

**S1 Fig. Frequency of the ancestral and derivative allele of rs1801279, rs1801280, rs1799929, rs1799930, rs1799931, rs1041983, and rs1208 in the African populations. Note:** A, Adenine; C, Cytosine, G, Guanine; T, Thymine.
(TIFF)

**S2 Fig. Frequency of the ancestral and derivative allele of rs1801279, rs1801280, rs1799929, rs1799930, rs1799931, rs1041983, and rs1208 in the Americas populations. Note:** A, Adenine; C, Cytosine, G, Guanine; T, Thymine.
(TIFF)

**S3 Fig. Frequency of the ancestral and derivative allele of rs1801279, rs1801280, rs1799929, rs1799930, rs1799931, rs1041983, and rs1208 in the Asian populations. Note:** A, Adenine; C, Cytosine, G, Guanine; T, Thymine.
(TIFF)

**S4 Fig. Frequency of the ancestral and derivative allele of rs1801279, rs1801280, rs1799929, rs1799930, rs1799931, rs1041983, and rs1208 in the European populations. Note:** A, Adenine; C, Cytosine, G, Guanine; T, Thymine.
(TIFF)

**S5 Fig. Frequency of the ancestral and derivative allele of rs1801279, rs1801280, rs1799929, rs1799930, rs1799931, rs1041983, and rs1208 in the Middle East populations. Note:** A, Adenine; C, Cytosine, G, Guanine; T, Thymine.
(TIFF)

**S6 Fig. Frequency of the most common haplotypes among the geographic regions.**
(TIF)

**S7 Fig. Box plots of the slow phenotype distribution among the Africa regions. Note:** CAf, Central Africa; EAf, East Africa; NAf, North Africa; SAf, South Africa; Waf, West Africa. CMR, Cameroon; ETH, Ethiopia; NAM, Namibia; NGA, Nigeria; TZA, Tanzania; ZAF, South

Africa.
(TIF)

**S8 Fig. Box plots of the slow phenotype distribution among the European regions. Note:** EEu, East Europe; Neu, North Europe; SEu, South Europe; WEu, West Europe. CZE, Czech Republic; FIN, Finland; FRA, France; GRC, Greece; ITA, Italy; NLD, the Netherlands; RUS, the Russian Federation; SRB, Serbia; SWE, Sweden.
(TIF)

**S9 Fig. Box plots of the slow phenotype distribution among the Americas regions. Note:** CAf, Central America; Nam, North America; Sam, South America. BRA, Brazil; COL, Colombia; ECU, Ecuador; MEX, Mexico; NIC, Nicaragua; PAN, Panama; USA, the United States of America.
(TIF)

**S10 Fig. Box plots of the slow phenotype distribution among the Asia regions. Note:** Cas. Central Asia; EAs, East Asia; SEAs, Southeast Asia; SAs, South Asia. JPN, Japan; KGZ, Kirghizstan; RUS, the Russian Federation; UZB, Uzbekistan.
(TIF)

**S11 Fig. Box plots of the fast phenotype distribution among the Asia regions. Note:** Cas. Central Asia; EAs, East Asia; SEAs, Southeast Asia; SAs, South Asia. JPN, Japan; KGZ, Kirghizstan; RUS, the Russian Federation; UZB, Uzbekistan.
(TIF)

**S12 Fig. Box plots of the fast phenotype distribution among the Americas regions. Note:** CAf, Central America; Nam, North America; Sam, South America. BRA, Brazil; COL, Colombia; ECU, Ecuador; NIC, Nicaragua; PAN, Panama; USA, the United States of America.
(TIF)

**S13 Fig. Box plots of the fast phenotype distribution among the Africa regions. Note:** CAf, Central Africa; EAf, East Africa; NAf, North Africa; SAf, South Africa; Waf, West Africa. CMR, Cameroon; ETH, Ethiopia; NAM, Namibia; NGA, Nigeria; TZA, Tanzania; ZAF, South Africa.
(TIF)

**S14 Fig. Box plots of the fast phenotype distribution among the European regions. Note:** EEu, East Europe; Neu, North Europe; SEu, South Europe; WEu, West Europe. CZE, Czech Republic; DEU, Germany; FRA, France; ITA, Italy; NLD, the Netherlands; RUS, the Russian Federation; SRB, Serbia.
(TIF)

**S1 Table. Data extraction of allele, genotype, haplotypes, and acetylator status from all articles included in this systematic review.**
(XLSX)

**S2 Table. Single nucleotide polymorphisms with nucleotide changes and phenotype of all haplotypes found in the present systematic review.**
(XLSX)

## Acknowledgments

We would like to thank Rosa del Carmen Milan Segovia, PhD and Lucia Taja-Chayeb, PhD for giving us free access to their data. We also thank to Opata Edward Kwame, M. Sc., for his

disinterested help in the proofreading. The authors would like to thank the anonymous reviewers; their suggestions remarkably increased the quality of our work.

## Author Contributions

**Conceptualization:** Jorge E. Gutiérrez-Virgen, Maricela Piña-Pozas, Rocío Gómez.

**Data curation:** Jorge E. Gutiérrez-Virgen, Esther A. Hernández-Tobías, Marco A. Meraz-Ríos, Rocío Gómez.

**Formal analysis:** Jorge E. Gutiérrez-Virgen, Esther A. Hernández-Tobías, Rocío Gómez.

**Investigation:** Jorge E. Gutiérrez-Virgen, Maricela Piña-Pozas, Lucia Taja-Chayeb, Ma. de Lourdes López-González, Marco A. Meraz-Ríos, Rocío Gómez.

**Methodology:** Jorge E. Gutiérrez-Virgen, Maricela Piña-Pozas, Esther A. Hernández-Tobías, Lucia Taja-Chayeb, Ma. de Lourdes López-González, Marco A. Meraz-Ríos, Rocío Gómez.

**Project administration:** Rocío Gómez.

**Resources:** Maricela Piña-Pozas, Rocío Gómez.

**Supervision:** Lucia Taja-Chayeb, Marco A. Meraz-Ríos, Rocío Gómez.

**Validation:** Rocío Gómez.

**Visualization:** Rocío Gómez.

**Writing – original draft:** Jorge E. Gutiérrez-Virgen, Rocío Gómez.

**Writing – review & editing:** Rocío Gómez.

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
