## [Decision Letter · Decision Letter 0]

9 Aug 2022

PONE-D-22-15782NAT2 global landscape: genetic diversity and acetylation statuses from a systematic reviewPLOS ONE

Dear Dr. GOMEZ,

Thank you for submitting your manuscript to PLOS ONE. After careful consideration, we feel that it has merit but does not fully meet PLOS ONE’s publication criteria as it currently stands. Therefore, we invite you to submit a revised version of the manuscript that addresses the points raised during the review process.

We look forward to receiving your revised manuscript.

Kind regards,

Qasim Ayub, Ph.D.

Academic Editor

PLOS ONE

Journal Requirements:

" Unfunded studies"

“The scholarship 1008579 was granted by the National Council of Science and Technology to J.E. 507 G-V. during his master’s in science studies.”

" Unfunded studies"

5. We note that Figures 8 to 16 in your submission contain map images which may be copyrighted. All PLOS content is published under the Creative Commons Attribution License (CC BY 4.0), which means that the manuscript, images, and Supporting Information files will be freely available online, and any third party is permitted to access, download, copy, distribute, and use these materials in any way, even commercially, with proper attribution. For these reasons, we cannot publish previously copyrighted maps or satellite images created using proprietary data, such as Google software (Google Maps, Street View, and Earth). For more information, see our copyright guidelines: http://journals.plos.org/plosone/s/licenses-and-copyright.

 a. You may seek permission from the original copyright holder of Figures 8 to 16 to publish the content specifically under the CC BY 4.0 license. 

Reviewers' comments:

Reviewer's Responses to Questions

**Comments to the Author**

1. Is the manuscript technically sound, and do the data support the conclusions?

Reviewer #1: Yes

Reviewer #2: Partly

2. Has the statistical analysis been performed appropriately and rigorously? 

Reviewer #1: Yes

Reviewer #2: No

3. Have the authors made all data underlying the findings in their manuscript fully available?

Reviewer #1: Yes

Reviewer #2: Yes

4. Is the manuscript presented in an intelligible fashion and written in standard English?

Reviewer #1: Yes

Reviewer #2: No

5. Review Comments to the Author

Reviewer #1: The authors presented the global landscape of genetic diversity and acetylation statuses of the NAT2 gene. Polymorphism in this gene is associated with cancer susceptibility and adverse side effects of the drugs. The importance of this gene in precision medicine and systematic analysis with a deep literature survey make this study interesting.

- The authors present a systematic review spanning the genetic and acetylation patterns from 167 articles obtained from October 1992 to October 2020. Why 1992 is chosen as starting point for literature searches.

- The authors did not include those papers in the literature survey for which full-text access was denied. Many good-quality journals including Nature Genetics, and Genome Research have restricted access to many of their online papers. How did the authors make sure that this exclusion did not affect the outcome of the analysis?

- The overall methodology is appropriate and effective.

- Do the inclusion criteria have a cut-off number of participant, below which for instance study is rejected being insignificant?

- Do the indels have any role in the phenotypic presentation of NAT2? There is no0mention of indels.

- Statements given in line number 156 i.e. “As such, only fast and slow acetylator status were reported.” and lines 161-162 “The different acetylator statuses among populations were separated into slow, intermediate, and fast acetylator, and they were depicted in a global map obtained from Mapswire are contradicting, in terms of inclusion of intermediate acetylator status.

- The studies carried out on Asian living in western countries pose another problem. They are increasingly inbred even more than their related populations in their native countries. Sometime these studies do not reflect the true picture of larger population. Did the authors scrutinize this fact? https://www.science.org/doi/10.1126/science.aac8624. The authors mentioned that 70% percent of Asian studies belongs to China and Japan and these populations have a different genetic structure than south Asian regions. This will apparently limit the wider applicability of results in Asian population.

-Americans are more diverse than with 8,551 haplotypes as compared to Asians 7,874. Given the huge population difference toward Asian continent how this is explained? Likewise, European population demonstrated highest haplotype diversity after global one. In terms of population genetic Americans and Europeans being on the other end of migration rout should have been less diverse. Same is describedin discussions Thus, Africans and Americans were the most diverse populations in regard to NAT2 diversity reinforcing the relevance of including ethnic minorities in

studies on diversity.

Reviewer #2: In this manuscript, the authors report on a systematic review of all published data on 7 polymorphisms in the coding exon, plus a regulatory variant, of the gene NAT2 in world-wide human populations. The authors screened more than 1000 publications so as to retrieve, after seemingly careful quality control, frequencies of NAT2 alleles (i.e. SNPs) and/or genotypes (at these SNPs), and/or haplotypes and/or phenotypes in 321 diverse human populations/groups. They then used this dataset to document the haplotypic and phenotypic diversity of NAT2 in humans.

I particularly appreciate the initiative and effort as we undertook a similar process some 15 years ago that gave rise to a publication that the authors cite (Sabbagh et al. 2011). Hence I believe that the study is valuable and I understand the amount of work devoted to it, but unfortunately the manuscript is not, in my view, ready for publication. Actually, I believe that it still needs substantial additional thinking and work.

The main reason for my reluctance to accept it in its present state is that, besides that the manuscript includes several vague assertions, the study seems to suffer from some methodological problems and even what I believe to be mistakes.

The first mistake, in my view, is that the authors do not take bias in SNP choice and level of resolution into proper consideration. They state that (Mat & Met section, I quote) “solely those phenotypes assigned from at least five SNPs were used”. Used for what, and which SNPs ? This is a central question and a main problem. Just as an example, Figure 8B displays the frequency of fast, intermediate and slow acetylator phenotypes in ARG-PRY, which according to the figure legend are Argentina and Paraguay, but if one looks at Supplementary Table S1, the data can only come from Bailliet et al. 2007, who only tested 3 SNPs (i.e. rs1799929/481, rs1799930/590 and rs1799931/857)… Thus SNP rs1801280/341, involved in the *5 haplotype determination (among others) which can lead to a slow or intermediate phenotype inference, was not tested in that study…

Another example of this problem of resolution level is provided in the Mat & Met section, where the authors state (I quote) “In those cases where the authors did not define the specific haplotype (i.e., NAT2*5A) and only reported the general haplotype (i.e., NAT2*5), the most frequent one was taken by default.” What does this mean ? If authors reported that they found haplotype *5 in their publication, and not *5A, it is probably because they hadn’t genotyped the marker that defines the *5A series (according to Supplementary Table S2, this would be rs1208, wouldn’t it ?). Thus using the most frequent one (most frequent where, by the way ?) can introduce quite some bias.

I believe that substantial standardization of the level of resolution is needed and should be well explained in the manuscript (or if needed in a Supplementary text file). I would advise the authors to start by deciding of a set of SNPs that have to be included for a sample to be considered, then infer the haplotype resolution possible for that set of SNPs, and infer the genotypes and phenotypes on that basis. If the authors still wish to IMPUTE some data (as they seem to have done with the *5/*5A example), then this should be clearly stated in Table S1 as well.

Linked to this, it is of no surprise, I believe, that recent studies (such as those on African or Middle Eastern populations for instance) reveal more diversity, and thus more singletons, than more ancient studies: the technology has quite improved in 30 years … from single SNPs genotyping to sequencing the whole gene (which by the way is a very small gene). By sequencing one avoids of course the SNP-choice bias, but this was not possible some years ago. Hence, limited sets of pre-selected SNP were genotyped in many early studies. This is not properly addressed in the analyses and in their interpretation. If one uses a limited set of SNPs, it is highly unlikely that one will find many new haplotypes (i.e. singletons).

Then, I found all the presentation on phenotypes very confusing. I believe that phenotypes were always inferred from haplotypes, is this right ? But then I did not understand how intermediate phenotypes were inferred ? I quote the authors “In the same way, the phenotype frequencies were reconstructed using the haplotypes reported by the authors following the criteria of the consensus nomenclature [https://nat.mbg.duth.gr]. As such, only fast and slow acetylator status were reported.” So, where are the intermediate acetylators ? The authors follow by stating that (I quote) “Nonetheless, these data were neither used in the network analysis nor the acetylation status frequencies.” So what data were used ? They continue in the next paragraph by stating that (I quote) “The different acetylator statuses among populations were separated into slow, intermediate, and fast acetylator ....” Which different acetylator statuses ? They follow (I quote) “Some authors reported the acetylator status as a fast-intermediate phenotype; these data were included in tables …” Which tables ? They follow (I quote) “… but were not used in depicting maps. In those articles where the acetylation status was not reported, it was inferred (i.e., fast and slow) from the haplotypes reported according to the consensus nomenclature [https://nat.mbg.duth.gr]. These data were not reflected on phenotype frequencies where only the acetylator status reported was considered;” … I am totally lost here. What was done exactly? When acetylator phenotype is reported in a paper, (whatever the classification used), one should check HOW these phenotypes were inferred (the method!). I would advise to add this information directly into Supplementary Table S1, so that one can immediately see in the Table for which population the phenotypes were inferred from haplo/diplotypes (and under which classification scheme: fast/slow, fast/intermediate/slow) and for which population they were actually measured. By the way, one has to consider also the method used to measure phenotypes, because this can influence the results… Moreover, and more generally, I failed to understand the difference between phenotype and acetylator status. What is the difference ?

Another major concern I have is related to the networks. The authors produced 7 networks of haplotypes, 1 global and 6 by geographic region (or sub-region/sub-groups, figures 3 to 7). I might be wrong here, but I believe that branch lengths in networks should be proportional to the number of mutational steps separating nodes (haplotypes) ? In those figures, this is not the case. For instance, in Figure 3, I see a branch linking *13A and *6F, with 3 mutational steps on this very short branch, whereas the very long branch linking *6A and *6J has only one mutational step indicated. Such a network is distorted, I believe, and does not properly represent what it should : the molecular diversity of haplotypes and the most parsimonious evolutionary steps linking those haplotypes. At present, these networks are unreadable, with many difficult to understand reticulations. Moreover, they are presented in such a way that one cannot directly compared them between the diverse world regions considered. For instance, Figure 3 has *5B on the right and *6A and the upper-left, whereas Figure 4 has *5B on the lower-left and *6A on the upper-right. I recommend some standardization here too. But most importantly, what are these networks used for ? At present, I only read some statements on the frequency of singletons… but without any mention to sample size. Identification of singletons is more probable as sample size increases. For instance, the authors state that (I quote) “Of note are the singletons and the regional haplotypes in Latinos and Native Americans (Figs 5B and 5C).”, and “The admixed populations from Brazil and Mexico exhibited many singletons and local haplotypes (*6J, *7C, and *7E) (Fig 5B)”. Indeed, in Figure 5B for instance, most singletons are from Brazil (and most of the haplotypic diversity indeed), not from Nicaragua and Peru. But look at the sample sizes in Table S1: Brazil 1323 individuals tested, against 137 in Nicaragua and 150 in Peru; 10 times more individuals tested ! Unless the authors included only subsets of those samples in some networks ? It seems so when one compares Figures 5A, 5B and 5C, because several Brazilian haplotypes of Fig 5B (supposedly South American in Fig 5A?) don’t show up in Fig 5C (e.g. *14A, *14E, *14F, just to name a few). I would advise that this information is clearly stated (e.g. one could imagine indicating in Table S1 in which figure(s) each sample is included).

My next point regards some clarification of the assignment of populations to groups. There is a whole paragraph listing those groups in the Mat & Met section (in the section for networks), but I see issues here. One is that the borders separating those groups are not indicated, so that, for instance, I don’t understand who the Asians are in the Asia and Pacific region (APA) group, as compared to the Northeast Asians (NEA) or Southeast Asians (SEA) ? The more so, as these subdivisions apparently don’t match those of Table S1 (in which a Central and Southern Asia group is included, and an Oceanian group constituted of a single Papuan sample, but nothing called APA…). Does this mean that Indians, Kirghiz and Tajiks were included in the same group as Papuans ? What would be the rationale for such a grouping ? The subdivision of American populations is even more difficult to follow, the more so as different subdivisions were used in different sections of the manuscript (e.g. in the Results section, the authors state (I quote) “The cosmopolitan populations from the USA and Canada provided 5,107 haplotypes”… in which group are these cosmopolitan populations included ? Hispanics (no code), Latinos (no code), NaM, or NHW ?). This is all very confusing and calls for standardization throughout the manuscript. Maybe that a map depicting the geographic groupings could be used ?

Then, there are also seemingly some misconceptions about populations in other world regions than America, such as (I quote) “Particularly, studies on African aboriginals have been associated with an interest in ….” What are African aboriginals ? Are Chinese Han also aboriginals ? Or Germans ? Many populations in sub-Saharan Africa trace back their origins to the spread of agriculture, just as Han Chinese (and probably that the population history is even more complex than just a link with the Neolithic, as is becoming more and more clear for Europe for instance). The authors should pay particular attention to such statements.

The central point of this manuscript, as I understand it, is to compare NAT2 genetic and phenotypic diversity across world populations. Hence, proper statistical testing of apparent differences should be systematically used instead of listing frequencies, or h and MPD in the text, and commenting without testing. The authors could consider making a table of h and MPD, which should also help them in commenting their results. For instance, they state that (I quote) “SSA was more diverse (h = 0.988 +- 0.001, MPD = 2.478 +- 1.339) than MEA (h = 0.946 +- 0.001, MPD = 2.588 +- 1.387)”. I agree that h is SLIGHTLY higher in SSA than in MEA (but is statistically significantly so ?), but MPD is SLIGHTLY higher in MEA than in SSA, so what ? There is also a whole paragraph on the comparison with the 1KG data that is vague at best. For instance, (I quote) “Of note is the behaviour of the derivate allele of rs1799931 in SAS populations; the frequency reported in this review was remarkably higher than those reported by 1KGP (Figs 9 - 15).” How much higher ? Such assertions (of which there are many instances of throughout the manuscript) should be grounded on numbers (and if possible, statistical tests of the differences..).

Related to this last point, I would also like to advise the authors to maybe read again our 2011 publication, in which we showed, as the authors stated in their abstract that (I quote) “NAT2*4, *5B, and *6A were the most frequent haplotypes globally. Nonetheless, the distribution of *5B and *7B were less and more frequent in Asians, respectively.” This is exactly what we had shown more than 10 years ago (see Sabbagh et al. 2011, Figure 2), which does not make the finding very new. And then, they continue with (I quote) “Regarding the acetylator status, East Asians and Native Americans harboured the highest frequencies of the fast phenotype, followed by South Europeans.” I do agree for many East Asian populations (which had already been shown in several studies in the mid-2000, of which also the Sabbagh et al. 2008 and 2011) and for some of the American populations, but South Europe, is this so ? It is rather SCE (South Central Europe according to Fig 8A) or the Swiss (CHE) according to Fig 8B, but Italians, Spanish and Turks don’t have such a high fast acetylation frequency seemingly. And what about the NAM (Namibians) in Fig 8B, or CMR (Cameroon). This again is all very confusing. I would advise the authors to read our review chapter on this topic (chapter 1.7) which was published in the book “Arylamine N-acetyltransferases in Health and Disease” (https://www.worldscientific.com/worldscibooks/10.1142/10763#t=aboutBook ). I can send them a PDF copy if they wish me to do so.

Then, except for Figure 11, all those figures showing the distribution of single SNPS are irrelevant, because they are not mutually independent. The information on single SNPs is included in the information on haplotypes. If Cambodians have a high frequency of rs1799931-A, it is due to the fact that they have a high frequency of haplotypes *7 (1 x *7A and 5 x *7B), but note the sample size: 7 individuals …

Finally, in my view, the lengthy Discussion section presents a lot of speculations but has little relation with the results. I would advise to rework it in a way much more focused on the results. The authors should take care of being much more clear in their statements. For instance, they discuss adaptation to fire use, but the use of fire for cooking is a cultural item that is shared among most (if not all) human populations, whether these are, still today, relying for their subsistence on hunting-and-gathering, or on food production. And one should not forget that before being food-producers, a rather recent cultural change in human history, all human populations were hunter-gatherers, whether in the Americas, in Asia, in Europe or in Africa. Maybe that the most interesting part on which the authors could focus might be the link between cultural practices in Asia and in America.

There are many other smaller points that need improvement, such as, if I understand properly Supplementary Table S1, phenotype frequencies could only be inferred from 285 samples, not 321, and unless I missed something, the table reports the frequencies of SNPs in 317 samples (not 321). Could the authors explain ? And what do the numbers on line 333 of Table S1 represent ? Or correct “Missence” to “Missense” in Table S1. Or what is the “ancestral burden” (abstract) ? Or which are “the controversial findings between acetylator states and the susceptibility to diseases” (abstract) ? But this is altogether already a very lengthy review.

I hope that all my remarks are constructive and will help the authors to substantially improve their manuscript. Again, I believe that the study is valuable, but it has to be consequently revised to get ready for publication.

Best regards,

Estella Poloni

6. PLOS authors have the option to publish the peer review history of their article (what does this mean?). If published, this will include your full peer review and any attached files.

Reviewer #1: **Yes: **Dr. Ishtiaq Ahmed Khan

Reviewer #2: **Yes: **Estella S. Poloni

---

## [Author Response · Author response to Decision Letter 0]

9 Feb 2023

Response to Reviewer Comments

First of all, we acknowledge the meticulous revision from the reviewers. We also thank the reviewers and the Editor for evaluating our manuscript. Overall, all sections have been reviewed and improved. We have carefully evaluated the points raised in the review and made corresponding changes to the text. These changes have significantly improved the quality of the manuscript. All changes appear highlighted in yellow (reviewer1) and in cyan (reviewer2) and in grey (for the Journal's requirements) in the track changes document. Our responses to the reviewer's remarks appear in blue font below. 

Reviewer #1. 

The authors presented the global landscape of genetic diversity and acetylation statuses of the NAT2 gene. Polymorphism in this gene is associated with cancer susceptibility and adverse side effects of the drugs. The importance of this gene in precision medicine and systematic analysis with a deep literature survey make this study interesting.

1. “The authors present a systematic review spanning the genetic and acetylation patterns from 167 articles obtained from October 1992 to October 2020. Why 1992 is chosen as starting point for literature searches”. 

Thank you for this concern. While the first papers regarding Nat2 were published around the 50' to 70's, these were related to isoniazid and tuberculosis. Nonetheless, using the search strategy described in our manuscript, we found at most ten papers prior to 1992, some of which were made in animal models. In turn, we considered using 1992 as a cut-off to include these papers in the systematic review.

2. “The authors did not include those papers in the literature survey for which full-text access was denied. Many good-quality journals including Nature Genetics, and Genome Research have restricted access to many of their online papers. How did the authors make sure that this exclusion did not affect the outcome of the analysis?” 

We apologize for this misunderstanding, and we appreciate your comment. The text was clarified in the Materials and Methods section (Eligibility criteria subsection). Briefly, we used several databases with which our institutions have agreements (e.g., Medline/Pubmed, Lilacs, Scielo, Conricyt). These databases include many prestigious journals, allowing us to include an enormous quantity of articles. Hence, we are certain that our exclusion criteria have not affected the outcome of our analyses, given that we were not able to find at most ten articles in full-text.

3. "The overall methodology is appropriate and effective”. 

Thank you so much for your appreciation. 

4. “Do the inclusion criteria have a cut-off number of participant, below which for instance study is rejected being insignificant?”

Many thanks for this question. We decide, by convenience, that all studies should have at least two individuals genotyped to be included. This information was incorporated in the Materials and Methods section (Eligibility criteria subsection).

5. “Do the indels have any role in the phenotypic presentation of NAT2? There is no mention of indels”. 

Thank you for this observation. We appreciate it. We have included this information in the Introduction section with its respective citations.

6. “Statements given in line number 156 i.e. “As such, only fast and slow acetylator status were reported.” and lines 161-162 “The different acetylator statuses among populations were separated into slow, intermediate, and fast acetylator, and they were depicted in a global map obtained from Mapswire are contradicting, in terms of inclusion of intermediate acetylator status”. 

We agree with your comment, and given the observations made by Reviewer 2 we have eliminated the statements among lines 154 to 157. This is because haplotype phenotype reconstruction could lack accuracy and detract from the actual outcomes.

7. “The studies carried out on Asian living in western countries pose another problem. They are increasingly inbred even more than their related populations in their native countries. Sometime these studies do not reflect the true picture of larger population. Did the authors scrutinize this fact? https://www.science.org/doi/10.1126/science.aac8624. The authors mentioned that 70% percent of Asian studies belongs to China and Japan and these populations have a different genetic structure than south Asian regions. This will apparently limit the wider applicability of results in Asian population”. 

We thank the reviewer for this remark. We agree with this observation that we had not considered. Your recommendations have been attended to in the Discussion section using the article suggested and others associated with this topic.

8. “Americans are more diverse than with 8,551 haplotypes as compared to Asians 7,874. Given the huge population difference toward Asian continent how this is explained? Likewise, European population demonstrated highest haplotype diversity after global one. In terms of population genetic Americans and Europeans being on the other end of migration rout should have been less diverse. Same is described in discussions Thus, Africans and Americans were the most diverse populations in regard to NAT2 diversity reinforcing the relevance of including ethnic minorities in studies on diversity”. 

We apologize for this misunderstanding, and many thanks for this suggestion. Firstly, the number of haplotypes you pointed out (i.e., 8,551 and 7,874) depends on the sample size. The misunderstanding was our mistake. In order to be more accurate, we have clarified your concerns in the Results description in all the sections. Indeed, the Americas presented 31 different haplotypes, whereas Asia only sixteen. About the differences in the diverse patterns (Asia versus the Americas as well as in the Americas versus Europe), we have included the answer to your questions in the Discussion section. 

Reviewer #2.

In this manuscript, the authors report on a systematic review of all published data on 7 polymorphisms in the coding exon, plus a regulatory variant, of the gene NAT2 in world-wide human populations. The authors screened more than 1000 publications so as to retrieve, after seemingly careful quality control, frequencies of NAT2 alleles (i.e. SNPs) and/or genotypes (at these SNPs), and/or haplotypes and/or phenotypes in 321 diverse human populations/groups. They then used this dataset to document the haplotypic and phenotypic diversity of NAT2 in humans. I particularly appreciate the initiative and effort as we undertook a similar process some 15 years ago that gave rise to a publication that the authors cite (Sabbagh et al. 2011). Hence, I believe that the study is valuable and I understand the amount of work devoted to it, but unfortunately the manuscript is not, in my view, ready for publication. Actually, I believe that it still needs substantial additional thinking and work. The main reason for my reluctance to accept it in its present state is that, besides that the manuscript includes several vague assertions, the study seems to suffer from some methodological problems and even what I believe to be mistakes.

1. The first mistake, in my view, is that the authors do not take bias in SNP choice and level of resolution into proper consideration. They state that (Mat & Met section, I quote) “solely those phenotypes assigned from at least five SNPs were used”. Used for what, and which SNPs ? This is a central question and a main problem. Just as an example, Figure 8B displays the frequency of fast, intermediate and slow acetylator phenotypes in ARG-PRY, which according to the figure legend are Argentina and Paraguay, but if one looks at Supplementary Table S1, the data can only come from Bailliet et al. 2007, who only tested 3 SNPs (i.e. rs1799929/481, rs1799930/590 and rs1799931/857). Thus, SNP rs1801280/341, involved in the *5 haplotype determination (among others) which can lead to a slow or intermediate phenotype inference, was not tested in that study. 

You are right. As you mentioned, our work was hard, and we made a mistake in including data obtained from fewer SNPs contravening our inclusion criteria. These data have been removed from maps but were maintained in Table S1. Likewise, we have clarified that only those acetylator statuses obtained from at least six SNPs were used for the analyses. These concerns were modified and clarified in the section Materials and Methods, reverberating in the Results section.

2. “Another example of this problem of resolution level is provided in the Mat & Met section, where the authors state (I quote) “In those cases where the authors did not define the specific haplotype (i.e., NAT2*5A) and only reported the general haplotype (i.e., NAT2*5), the most frequent one was taken by default.” What does this mean ? If authors reported that they found haplotype *5 in their publication, and not *5A, it is probably because they hadn’t genotyped the marker that defines the *5A series (according to Supplementary Table S2, this would be rs1208, wouldn’t it ?). Thus using the most frequent one (most frequent where, by the way ?) can introduce quite some bias. I believe that substantial standardization of the level of resolution is needed and should be well explained in the manuscript (or if needed in a Supplementary text file). I would advise the authors to start by deciding of a set of SNPs that have to be included for a sample to be considered, then infer the haplotype resolution possible for that set of SNPs, and infer the genotypes and phenotypes on that basis. If the authors still wish to IMPUTE some data (as they seem to have done with the *5/*5A example), then this should be clearly stated in Table S1 as well”.

We agree, and we thank you for this observation. We have modified these sentences, and the SNPs considered for acetylator statuses, haplotypes, and network analyses have been defined in the Materials and Methods section. We hope your concern has been clarified with these modifications.

3. “Linked to this, it is of no surprise, I believe, that recent studies (such as those on African or Middle Eastern populations for instance) reveal more diversity, and thus more singletons, than more ancient studies: the technology has quite improved in 30 years … from single SNPs genotyping to sequencing the whole gene (which by the way is a very small gene). By sequencing one avoids of course the SNP-choice bias, but this was not possible some years ago. Hence, limited sets of pre-selected SNP were genotyped in many early studies. This is not properly addressed in the analyses and in their interpretation. If one uses a limited set of SNPs, it is highly unlikely that one will find many new haplotypes (i.e. singletons)”. 

Thank you for this comment. We have replicated your observation in the Discussion section.

4. “Then, I found all the presentation on phenotypes very confusing. I believe that phenotypes were always inferred from haplotypes, is this right? But then I did not understand how intermediate phenotypes were inferred I quote the authors “In the same way, the phenotype frequencies were reconstructed using the haplotypes reported by the authors following the criteria of the consensus nomenclature [https://nat.mbg.duth.gr]. As such, only fast and slow acetylator status were reported.” So, where are the intermediate acetylators? The authors follow by stating that (I quote) “Nonetheless, these data were neither used in the network analysis nor the acetylation status frequencies.” So what data were used ? They continue in the next paragraph by stating that (I quote) “The different acetylator statuses among populations were separated into slow, intermediate, and fast acetylator ....” Which different acetylator statuses ? They follow (I quote) “Some authors reported the acetylator status as a fast-intermediate phenotype; these data were included in tables …” Which tables ? They follow (I quote) “… but were not used in depicting maps”. 

Thank you for your exhaustive review. It was very helpful in revising and improving our paper. All these mistakes have been replied to in the new version of the Material and Methods section. In addition, we have meticulously reviewed the Table exhibiting these data. In this new version, all data are homogeneous, and we have modified all mistakes. 

5. “In those articles where the acetylation status was not reported, it was inferred (i.e., fast and slow) from the haplotypes reported according to the consensus nomenclature [https://nat.mbg.duth.gr]. These data were not reflected on phenotype frequencies where only the acetylator status reported was considered;” … I am totally lost here. What was done exactly? When acetylator phenotype is reported in a paper, (whatever the classification used), one should check HOW these phenotypes were inferred (the method!). I would advise to add this information directly into Supplementary Table S1, so that one can immediately see in the Table for which population the phenotypes were inferred from haplo/diplotypes (and under which classification scheme: fast/slow, fast/intermediate/slow) and for which population they were actually measured. By the way, one has to consider also the method used to measure phenotypes, because this can influence the results. Moreover, and more generally, I failed to understand the difference between phenotype and acetylator status. What is the difference?”

We agree with these suggestions. Thus, we have clarified these paragraphs in the Material and Methods section.

6. “Another major concern I have is related to the networks. The authors produced 7 networks of haplotypes, 1 global and 6 by geographic region (or sub-region/sub-groups, figures 3 to 7). I might be wrong here, but I believe that branch lengths in networks should be proportional to the number of mutational steps separating nodes (haplotypes) ? In those figures, this is not the case. For instance, in Figure 3, I see a branch linking *13A and *6F, with 3 mutational steps on this very short branch, whereas the very long branch linking *6A and *6J has only one mutational step indicated. Such a network is distorted, I believe, and does not properly represent what it should : the molecular diversity of haplotypes and the most parsimonious evolutionary steps linking those haplotypes. At present, these networks are unreadable, with many difficult to understand reticulations. Moreover, they are presented in such a way that one cannot directly compared them between the diverse world regions considered. For instance, Figure 3 has *5B on the right and *6A and the upper-left, whereas Figure 4 has *5B on the lower-left and *6A on the upper-right. I recommend some standardization here too. But most importantly, what are these networks used for ? At present, I only read some statements on the frequency of singletons… but without any mention to sample size. Identification of singletons is more probable as sample size increases. For instance, the authors state that (I quote) “Of note are the singletons and the regional haplotypes in Latinos and Native Americans (Figs 5B and 5C).”, and “The admixed populations from Brazil and Mexico exhibited many singletons and local haplotypes (*6J, *7C, and *7E) (Fig 5B)”. Indeed, in Figure 5B for instance, most singletons are from Brazil (and most of the haplotypic diversity indeed), not from Nicaragua and Peru. But look at the sample sizes in Table S1: Brazil 1323 individuals tested, against 137 in Nicaragua and 150 in Peru; 10 times more individuals tested ! Unless the authors included only subsets of those samples in some networks ? It seems so when one compares Figures 5A, 5B and 5C, because several Brazilian haplotypes of Fig 5B (supposedly South American in Fig 5A?) don’t show up in Fig 5C (e.g. *14A, *14E, *14F, just to name a few). I would advise that this information is clearly stated (e.g. one could imagine indicating in Table S1 in which figure(s) each sample is included)”. 

We agree with all your comments. In this new version, we have modified the presentation of our data, including statistical analyses to detect the differences among populations. 

7. “My next point regards some clarification of the assignment of populations to groups. There is a whole paragraph listing those groups in the Mat & Met section (in the section for networks), but I see issues here. One is that the borders separating those groups are not indicated, so that, for instance, I don’t understand who the Asians are in the Asia and Pacific region (APA) group, as compared to the Northeast Asians (NEA) or Southeast Asians (SEA) ? The more so, as these subdivisions apparently don’t match those of Table S1 (in which a Central and Southern Asia group is included, and an Oceanian group constituted of a single Papuan sample, but nothing called APA…). Does this mean that Indians, Kirghiz and Tajiks were included in the same group as Papuans? What would be the rationale for such a grouping?” 

Thank you so much for this comment. We have used the continent subdivision proposed by the World Atlas webpage. In this setting, we considered that this subdivision could be better for our comparisons. This information appears in the Material and Methods section.

8. “The subdivision of American populations is even more difficult to follow, the more so as different subdivisions were used in different sections of the manuscript (e.g. in the Results section, the authors state (I quote) “The cosmopolitan populations from the USA and Canada provided 5,107 haplotypes”… in which group are these cosmopolitan populations included ? Hispanics (no code), Latinos (no code), NaM, or NHW ?). This is all very confusing and calls for standardization throughout the manuscript. Maybe that a map depicting the geographic groupings could be used ?”

Your comment is pertinent. We have clarified such ancestry subdivisions in the Material and Methods section.

9. “Then, there are also seemingly some misconceptions about populations in other world regions than America, such as (I quote) “Particularly, studies on African aboriginals have been associated with an interest in ….” What are African aboriginals ? Are Chinese Han also aboriginals ? Or Germans ? Many populations in sub-Saharan Africa trace back their origins to the spread of agriculture, just as Han Chinese (and probably that the population history is even more complex than just a link with the Neolithic, as is becoming more and more clear for Europe for instance). The authors should pay particular attention to such statements”. 

Many thanks for these suggestions. We have been more careful with the use of adjectives.

10. “The central point of this manuscript, as I understand it, is to compare NAT2 genetic and phenotypic diversity across world populations. Hence, proper statistical testing of apparent differences should be systematically used instead of listing frequencies, or h and MPD in the text, and commenting without testing. The authors could consider making a table of h and MPD, which should also help them in commenting their results. For instance, they state that (I quote) “SSA was more diverse (h = 0.988 +- 0.001, MPD = 2.478 +- 1.339) than MEA (h = 0.946 +- 0.001, MPD = 2.588 +- 1.387)”. I agree that h is SLIGHTLY higher in SSA than in MEA (but is statistically significantly so ?), but MPD is SLIGHTLY higher in MEA than in SSA, so what ? There is also a whole paragraph on the comparison with the 1KG data that is vague at best. For instance, (I quote) “Of note is the behaviour of the derivate allele of rs1799931 in SAS populations; the frequency reported in this review was remarkably higher than those reported by 1KGP (Figs 9 - 15).” How much higher ? Such assertions (of which there are many instances of throughout the manuscript) should be grounded on numbers (and if possible, statistical tests of the differences)”.

We acknowledge your comment. We have included a table (Table 1) with this data as well as the statistical analyses.

11. “Related to this last point, I would also like to advise the authors to maybe read again our 2011 publication, in which we showed, as the authors stated in their abstract that (I quote) “NAT2*4, *5B, and *6A were the most frequent haplotypes globally. Nonetheless, the distribution of *5B and *7B were less and more frequent in Asians, respectively.” This is exactly what we had shown more than 10 years ago (see Sabbagh et al. 2011, Figure 2), which does not make the finding very new. And then, they continue with (I quote) “Regarding the acetylator status, East Asians and Native Americans harboured the highest frequencies of the fast phenotype, followed by South Europeans.” I do agree for many East Asian populations (which had already been shown in several studies in the mid-2000, of which also the Sabbagh et al. 2008 and 2011) and for some of the American populations, but South Europe, is this so ? It is rather SCE (South Central Europe according to Fig 8A) or the Swiss (CHE) according to Fig 8B, but Italians, Spanish and Turks don’t have such a high fast acetylation frequency seemingly. And what about the NAM (Namibians) in Fig 8B, or CMR (Cameroon). This again is all very confusing. I would advise the authors to read our review chapter on this topic (chapter 1.7) which was published in the book “Arylamine N-acetyltransferases in Health and Disease” (https://www.worldscientific.com/worldscibooks/10.1142/10763#t=aboutBook ). I can send them a PDF copy if they wish me to do so”. 

We agree with this comment. In order to follow your suggestion, we have modified it. Thank you so much for giving us free access to this chapter.

12. “Then, except for Figure 11, all those figures showing the distribution of single SNPS are irrelevant, because they are not mutually independent. The information on single SNPs is included in the information on haplotypes. If Cambodians have a high frequency of rs1799931-A, it is due to the fact that they have a high frequency of haplotypes *7 (1 x *7A and 5 x *7B), but note the sample size: 7 individuals”.

Thank you so much for this suggestion. Nonetheless, we considered that the information provided by a figure is valuable and could give a general idea about the frequency distribution. We have modified the presentation of the frequencies using area plots, and we have included this information in the Supplemental Information section. 

13. “Finally, in my view, the lengthy Discussion section presents a lot of speculations but has little relation with the results. I would advise to rework it in a way much more focused on the results. The authors should take care of being much more clear in their statements. For instance, they discuss adaptation to fire use, but the use of fire for cooking is a cultural item that is shared among most (if not all) human populations, whether these are, still today, relying for their subsistence on hunting-and-gathering, or on food production. And one should not forget that before being food-producers, a rather recent cultural change in human history, all human populations were hunter-gatherers, whether in the Americas, in Asia, in Europe or in Africa. Maybe that the most interesting part on which the authors could focus might be the link between cultural practices in Asia and in America”.

Many thanks for this comment. We have modified it following your suggestions.

14. “There are many other smaller points that need improvement, such as, if I understand properly Supplementary Table S1, phenotype frequencies could only be inferred from 285 samples, not 321, and unless I missed something, the table reports the frequencies of SNPs in 317 samples (not 321). Could the authors explain ? And what do the numbers on line 333 of Table S1 represent ? Or correct “Missence” to “Missense” in Table S1. Or what is the “ancestral burden” (abstract) ? Or which are “the controversial findings between acetylator states and the susceptibility to diseases” (abstract) ? But this is altogether already a very lengthy review”. 

Thank you so much for this observation. The sentence was reformulated.

I hope that all my remarks are constructive and will help the authors to substantially improve their manuscript. Again, I believe that the study is valuable, but it has to be consequently revised to get ready for publication.

We agree with your suggestions. We attended them because we are sure that your expertise has improved our manuscript's quality and information. Thank you so much!

---

## [Decision Letter · Decision Letter 1]

16 Mar 2023

NAT2 global landscape: genetic diversity and acetylation statuses from a systematic review

PONE-D-22-15782R1

Dear Dr. GOMEZ,

We’re pleased to inform you that your manuscript has been judged scientifically suitable for publication and will be formally accepted for publication once it meets all outstanding technical requirements.

Kind regards,

Qasim Ayub, Ph.D.

Academic Editor

PLOS ONE

Additional Editor Comments (optional):

Reviewers' comments:

Reviewer's Responses to Questions

**Comments to the Author**

1. If the authors have adequately addressed your comments raised in a previous round of review and you feel that this manuscript is now acceptable for publication, you may indicate that here to bypass the “Comments to the Author” section, enter your conflict of interest statement in the “Confidential to Editor” section, and submit your "Accept" recommendation.

Reviewer #2: All comments have been addressed

2. Is the manuscript technically sound, and do the data support the conclusions?

Reviewer #2: (No Response)

3. Has the statistical analysis been performed appropriately and rigorously? 

Reviewer #2: (No Response)

4. Have the authors made all data underlying the findings in their manuscript fully available?

Reviewer #2: (No Response)

5. Is the manuscript presented in an intelligible fashion and written in standard English?

Reviewer #2: (No Response)

6. Review Comments to the Author

Reviewer #2: (No Response)

7. PLOS authors have the option to publish the peer review history of their article (what does this mean?). If published, this will include your full peer review and any attached files.

Reviewer #2: No

---

## [Editor Report · Acceptance letter]

21 Mar 2023

PONE-D-22-15782R1 

*NAT2* global landscape: genetic diversity and acetylation statuses from a systematic review 

Dear Dr. Gómez:

I'm pleased to inform you that your manuscript has been deemed suitable for publication in PLOS ONE. Congratulations! Your manuscript is now with our production department. 

Kind regards, 

on behalf of

Dr. Qasim Ayub 

Academic Editor

PLOS ONE